



# Measurement report: Ice nucleating abilities of biomass burning, African dust, and sea spray aerosol particles over the Yucatan Peninsula

Fernanda Córdoba[1,2,*], Carolina Ramirez-Romero[1,3,*], Diego Cabrera[1], Graciela B. Raga[1], Javier Miranda[4], Harry Alvarez Ospina[5], Daniel Rosas[6], Bernardo Figueroa[7], Jong Sung Kim[8], Jacqueline Yakobi-Hancock[8], Talib Amador[6], Wilfrido Gutierrez[1, †], Manuel García[1], Allan K. Bertram[9], Darrel Baumgardner[10], and Luis A. Ladino[1,**]

[1]Centro de Ciencias de la Atmósfera, Universidad Nacional Autónoma de México, Mexico City, Mexico
[2]Posgrado en Ciencias Química, Universidad Nacional Autónoma de México, Mexico City, Mexico
[3]Posgrado en Ciencias de la Tierra, Universidad Nacional Autónoma de México, Mexico City, Mexico
[4]Instituto de Física, Universidad Nacional Autónoma de México, Mexico City, Mexico
[5]Facultad de Ciencias, Universidad Nacional Autónoma de México, Mexico City, Mexico
[6]Facultad de Química, Universidad Autónoma de Yucatán, Mérida, Mexico
[7]Instituto de Ingeniería, Unidad Académica de Sisal, Universidad Nacional Autónoma de México, Sisal, Mexico
[8]Dalhousie University, Halifax, Nova Scotia, Canada
[9]Chemistry Department, University of British Columbia, Vancouver, Canada
[10]Droplet Measurement Technologies, Colorado, US
[†] Deceased
[*]These authors contributed equally to this work
[**]Correspondence to: Luis A. Ladino (luis.ladino@atmosfera.unam.mx)

**Abstract.** Most precipitation from deep clouds over the continents and in the intertropical convergence zone is strongly influenced by the presence of ice crystals, whose formation requires the presence of ice nucleating particles (INP). Although there are a large number of INP sources, the ice nucleating abilities of aerosol particles emitted from oceans, deserts, and wildfires are poorly described at tropical latitudes. To fill this gap in knowledge, the UNAM-MicroOrifice Uniform Deposit Impactor-Droplet Freezing Technique (UNAM-MOUDI-DFT) was built. Aerosol samples were collected in Sisal and Merida, Yucatan (Mexico) under the influence of cold fronts, biomass burning (BB), and African dust (AD), during five short-term field campaigns between January 2017 and July 2018.

The three different aerosol types were distinguished by characterizing their physicochemical properties. Marine aerosol (MA), BB, and AD air masses were found to contain INP; the





highest concentrations were found for AD (from 0.071 L$^{-1}$ to 36.07 L$^{-1}$), followed by MA

(from 0.068 L$^{-1}$ to 18.90 L$^{-1}$), and BB (from 0.063 L$^{-1}$ to 10.21 L$^{-1}$). However, MA had the

highest surface active site density ($n_s$) between -15°C and -30°C. Additionally, supermicron

particles contributed more than 72% of the total INP concentration independent of aerosol

type; MA had the largest contribution from supermicron particles.

## 1. Introduction


About 60% of the Earth's surface is covered by clouds at any time. However, the lack of

detailed knowledge about the aerosol-cloud interactions introduces large uncertainties in

projections of climate change (Lohmann and Feichter, 2005; Boucher et al., 2013). Globally,

more than 50% of the precipitation is initiated via the ice phase, and therefore, it is very

important to fully understand the formation of ice in clouds (Mülmenstädt et al., 2015).

Primary ice particle formation takes place in the atmosphere via homogeneous and

heterogeneous ice nucleation. Homogeneous ice nucleation typically occurs at temperatures

below -38°C and relative humidities with respect to ice (RH$_i$) above 140% (Knopf et al.,

2011; Kanji et al., 2017). In contrast, heterogeneous ice nucleation where ice nucleating

particles (INP) promote phase transition, occurs at temperatures warmer than -38°C and

lower RH$_i$ than those required for homogeneous nucleation (Kanji et al., 2017). Although

there are several pathways through which ice particles can form heterogeneously (i.e.,

deposition nucleation, pore-condensation freezing, immersion freezing, condensation

freezing, and contact freezing), immersion freezing has been reported as the most important

ice nucleation mode for mixed-phase clouds (Knopf et al., 2011; Murray et al., 2012).

The presence of INP in clouds affects its microphysical properties, cloud lifetime,

precipitation formation, and the planetary radiative balance (DeMott et al., 2003; DeMott et

al., 2010; Boucher et al., 2013; Kanji et al., 2017). Only 1 in 10$^5$ to 10$^6$ of the aerosol particles

can act as INP at temperatures higher than -38°C (Lohmann et al., 2016). Although it is not

completely clear which physicochemical properties make an aerosol particle a good INP, its

size, chemical composition, and the presence of active sites have been reported as key factors

(Pruppacher and Klett, 1997; Kanji et al., 2017). As summarized by Kanji et al. (2017),

different aerosol particles such as mineral and desert dust, crystalline salts, volcanic ash,



organic and glassy particles, metallic particles, biomass and fossil combustion particles, marine aerosol, and biological particles, have all been shown to act as INP under different thermodynamic conditions as a function of the ice nucleation mode. A short summary of the main results of the ice nucleating abilities found for mineral dust, biomass burning, and marine aerosol particles in the immersion freezing mode is provided below.


The ice nucleating abilities of different pure minerals, clays, and surrogates of natural mineral dust particles have been evaluated in the immersion freezing mode (e.g., Marcolli et al. 2007; Augustin-Bauditz et al. 2008; Lüönd et al. 2010; Murray et al. 2011; Wheeler et al. 2015; Harrison et al. 2019). These studies found that mineral dust particles are able to catalyze ice

formation at temperatures as high as -15.2°C. Boose et al. (2016) analyzed 15 ambient mineral dust samples, including some collected after they were transported from far away sources. The authors reported that the high ice nucleating abilities on those samples were related to the presence of K-feldspar, in agreement with earlier results reported by Atkinson et al. (2013). Boose et al. (2016) also show that airborne dust samples transported from far

away sources had lower ice nucleation activity, likely due to particle aging. Ardon-Dryer and Levin (2014) evaluated the ice nucleating abilities of mineral dust from the eastern Mediterranean region via immersion freezing, and found that INP concentrations varied between 0.16 L$^{-1}$ and 234 L$^{-1}$ at temperatures between -11.8°C and -28.9°C. Reicher et al. (2019) found similar INP concentration (from 0.1 L$^{-1}$ to $10^3$ L$^{-1}$) in the same region, during

dust storms, at temperatures ranging from -18.2°C to -38.2°C. Price et al. (2018) reported INP concentrations from 0.1 L$^{-1}$ to $10^2$ L$^{-1}$ at -12°C and -23°C under dusty conditions in Cape Verde, Africa, while Gong et al. (2020) found INP concentrations between $2x10^{-4}$ L$^{-1}$ and $2x10^{-1}$ L$^{-1}$ at temperatures between -5°C and -24°C, under low dust emissions in the same region.


Nine of the 21 different biomass fuels tested by Petters et al. (2009) produced INP, with swamp sawgrass smoke identified as the most efficient fuel reporting an INP fraction of ~1:100 particles. Umo et al. (2015) evaluated the ice-nucleating activity of coal fly ash (CFA), coal bottom ash, domestic bottom ash, and wood bottom. CFA was found to be the

most efficient, with a $T_{50}$ (the temperature at which 50% of the droplets freeze) around -16°C.



The authors identified particle morphology and chemical composition as the key physicochemical properties controlling the ice nucleating abilities of the studied biomass burning (BB) particles. In controlled laboratory experiments, Levin et al. (2016) analyzed 22 different biomass fuels and found that the INP concentrations ranged between $10^2$ L$^{-1}$ and

$10^4$ L$^{-1}$ at -30°C. They also observed increased INP concentrations for highly-efficient combustion (i.e., > 0.95). Prenni et al. (2012) found that INP concentrations from wildfires and prescribed fires in Colorado and Wyoming (United States) ranged between 3.4 L$^{-1}$ and 90 L$^{-1}$ at -30°C. Likewise, Mccluskey et al. (2014) found that mean INP concentrations during prescribed burns varied from 6.36 L$^{-1}$ to 16.7 L$^{-1}$ at -30°C in Colorado, and from 0.64 L$^{-1}$ to

34.03 L$^{-1}$ between -22°C to -30°C in Georgia. However, Mccluskey et al. (2014) showed that INP concentrations were higher during wildfires in Colorado (i.e., Hewlett Wildfire and High Park Wildfire), with mean values of 15.90 L$^{-1}$ to 79.43 L$^{-1}$ (between -22°C and -30°C) and 7.18 L$^{-1}$ to 75.98 L$^{-1}$ (between -22°C to -31°C).

Several field campaigns have been conducted to evaluate the ice nucleating abilities of aerosol particles in marine environments. Bigg (1973) collected aerosol samples in Australia, Schnell and Vali (1975) in Huntington Beach (California, US), in the Caribbean off Nassau (Bahamas), Vancouver (Canada), and Nova Scotia (Canada), Rosinski et al. (1987) over the Pacific Ocean, and Rosinski et al. (1988) in the Gulf of Mexico (GoM). More recently, the

ice nucleating abilities of the particles present in the Arctic the sea surface microlayer (Wilson et al., 2015; Irish et al., 2017; Irish et al., 2019) and the Arctic ambient aerosol were evaluated (DeMott et al., 2016; Creamean et al., 2018). Similar studies were performed over the eastern Mediterranean by Gong et al., (2019), over the Southern Ocean by McCluskey et al. (2018) and Welti et al. (2020), over the north Atlantic by Wilbourn et al. (2020), and over

the tropical Atlantic by DeMott et al. (2016), Welti et al. (2018), Ladino et al. (2019), Ladino et al. (2020), and Gong et al. (2020). Welti el al. (2020) showed INP concentrations for different zones in the Artic, Atlantic, Pacific, and Southern Ocean. In the N-Polar region they report INP concentrations ranging from $10^{-3}$ L$^{-1}$ to $10^{-1}$ L$^{-1}$ at temperatures from -7°C to -28°C, and $10^{-1}$ L$^{-1}$ to $10^6$ L$^{-1}$ at temperatures between -26°C to -38°C. For the N-Temperate

zone, the INP concentration was found to be similar than the N-Polar region at temperatures from -5°C to -25°C; however, at temperatures between -32°C to -38°C the INP concentration





was reported to vary between $10^{-1}$ L$^{-1}$ to $10^5$ L$^{-1}$. In the S-Temperate region, they found that the INP concentrations ranged between $10^{-3}$ L$^{-1}$ and $10^1$ L$^{-1}$ at temperatures from -5°C to -30°C, and $10^4$ L$^{-1}$ to $10^5$ L$^{-1}$ at temperatures between -35 L$^{-1}$ and -38°C. At the S-Polar zone, Welti el al. (2020) reported that the INP concentration ranged from $10^{-3}$ L$^{-1}$ to $10^{-1}$ L$^{-1}$ at temperatures from -5°C to -29°C. Finally, at tropical latitudes, INP concentrations were found to vary between $10^{-3}$ L$^{-1}$ to $10^{-1}$ L$^{-1}$ at temperatures from -5°C to -25°C, and $10^{-1}$ L$^{-1}$ to $10^5$ L$^{-1}$ at lower temperatures (i.e., -24°C and -38°C). It is important to note that in some of the aforementioned studies, onset freezing temperatures as high as -3°C were reported, which were associated with marine organic material, likely of biological origin.

Aerosol particles can be transported over long distances far away from their emission sources (e.g., Griffin et al., 2001; Taylor, 2002; Wu et al., 2004; Prenni et al., 2009). For example, Prospero and Lamb (2003), Prospero and Mayol-Bracero (2013), and Ramírez Romero et al., (2020) showed how mineral dust particles from the Saharan desert could reach the Caribbean region, the United States, and Mexico. The highest probability for mineral dust to reach the Caribbean and Mexico occurs during the mid-summer drought (MSD), a relative minimum in precipitation typically between mid-July and mid-August. Similarly, Peppler et al. (2000) and Saide et al. (2015) showed how BB particles from Central America and southern Mexico could impact the Yucatan Peninsula, Mexico. As shown by Ríos and Raga (2018), there is a clear BB seasonality in Mexico and Central America from December to June, with the largest burned areas between April and May. Finally, the arrival of cold fronts to southern Mexico has also been documented (DiMego et al., 1976; Cavazos, 1997). Air masses behind cold fronts, characterized by low temperatures, dry conditions, and strong winds, flowing over the GoM, bring marine aerosol particles into the Yucatan Peninsula, as shown by Ladino et al. (2019).

In summary, most of the field studies related to INP have been carried out in mid- and high-latitudes, with a limited amount of studies in tropical latitudes. As the tropics have significantly different characteristics than higher latitudes (e.g., flora, fauna, marine biological activity, sea surface temperature, air temperature, relative humidity (RH), cloud cover, atmospheric circulation, among others), there is an urgent need to improve the current





understating that tropical aerosol particles play in cloud formation (Yakobi-Hancok et al., 2014). Given that Mexico is yearly impacted by aerosol particles transported from Africa,

North America, and Central America, the present study evaluates the ice nucleating abilities of three different types of aerosol particles.

## 2. Methodology
### 2.1. Sampling location and methods

Five short-term field campaigns were carried out between January 2017 and July 2018, three in Merida (20.98°N, 89.64°W) and two in Sisal (21.16°N, 90.04°W) as part of the African Dust and Biomass Burning Over Yucatan (ADABBOY) project. Both sites are located in the Yucatan Peninsula (Figure 1) in southeastern Mexico. The Yucatan Peninsula is surrounded by the GoM in the North and West, the Caribbean Sea in the East, and Central America in

the South. Most of the Peninsula (i.e., 86%) presents a warm sub-wet climate, while the remaining 14% has dry and semi-dry climate. The annual average temperature and RH are 26°C and 79%, respectively (INEGI, 2019). Its soil is karstic, where the principal components are limestone, dolomite, and gypsum (Herrera et al., 2004).

Merida, the capital of Yucatan State located about 35 km off the coast of the GoM, has a population of 892,363 (INEGI, 2015). The main activities in town are tourism, manufacturing and textile industry, and commerce (Fuentes and Morales, 2000). The city has an average temperature of 25°C and an annual accumulated precipitation of about 900 mm (INEGI, 2017). The sampling site in Merida was located on the rooftop of the School of Chemistry of

the Universidad Autónoma de Yucatán (SC-UADY), located in the central-western part of the city and 3 km away from downtown.

Sisal is located 49 km from Merida and has 1,837 inhabitants (INEGI, 2010). It is a small coastal village where most people work in fishing activities without nearby industrial

activities (Santoyo, 2017). The mean temperature varies between 20°C and 34°C, and the RH is 80.8% ± 26.8%. Sisal is affected yearly by cold fronts between November and January and by convective storms and precipitation from the end of April to October (Santoyo, 2017). The aerosol sampling took place at the rooftop of the building of the Engineering Institute of





the Universidad Nacional Autónoma de México (EI-UNAM), which is 50 m away from the

shoreline. The dates of the sampling periods for Merida and Sisal are shown in Table 1. A

subset of the samples collected during the five campaigns was chosen for presented in this

study, as summarized in Table S1.

### 2.2. Instrumentation
### 2.2.1.   Ice-nucleating particles

The evaluation of the ice-nucleating abilities of the ambient aerosol particles consist of three

steps: i) collection of aerosol particles, ii) ice nucleation efficiency with the droplet freezing

technique (DFT), and iii) calculation of the INP concentration as a function of temperature

and particle size.


*Collection of aerosol particles*

Aerosol particles were collected by inertial impaction on hydrophobic glass coverslips (HR3-

215; Hampton Research) using a micro-orifice uniform deposit impactor (MOUDI 100R,

MSP), as shown in Figure 2a. To keep the glass coverslips on the MOUDI stages, substrate

holders were used as reported by Mason et al. (2015a). The MOUDI used has eight stages to

separate particles as a function of their aerodynamic diameter (cut sizes of 0.18, 0.32, 0.56,

1.0, 1.8, 3.2, 5.6, and 10.0 μm). The flow rate was set to 30.0 L min$^{-1}$ for the five campaigns;

except for Sisal 2018, with a flow rate of 25.8 L min$^{-1}$. The sampling time was typically 6 h

from 08:00 to 14:00 (local time); however, in some specific days, more than one sample per

day was collected. After each sampling, the glass coverslips were stored in petri dishes at

4°C prior to their analysis with the DFT in Mexico City (Fig. 2a).

*The UNAM-DFT*

The DFT has been employed in different studies to evaluate the ability of aerosol particles to

act as INP via immersion freezing (e.g., Koop et al., 2000; Iannone et al., 2011; Mason et al.,

2015a; Mason et al., 2015b; Wheeler et al., 2015; Si et al., 2018; Irish et al., 2019). A DFT

was built in the Micro and Mesoscale Interactions Laboratory of the Atmospheric Science

Center at the UNAM based on the design described in Mason et al.  (2015a). It consists of

four different parts/sections as follows: (i) cold stage, (ii) humid/dry air system, (iii) optical



microscope with video recording system, and (iv) data acquisition system, as shown in Figure
3.

The cold stage is integrated by a sample holder, a cooling block, and a heating block with all
three aligned on top of each other with the sample holder at the top and the cooling block at
the bottom (Fig. S1a). The cooling block is made of stainless steel (4x4x2 cm), where a
cooling liquid continuously circulates through it to maintain a constant temperature. A copper
heating block (4x4x1 cm) is located above the cooling block. Two resistances heaters (100
W and 120 V) are placed in the middle of the heating block to increase its temperature to
control the temperature of the sample holder. The temperature of the heating block is
regulated by a temperature controller via a thermocouple. On top of the heating block, a
sample holder (4x2x1 cm) made of stainless steel is placed. To ensure good thermal contact
between the sample holder, the heating block, and the cold block, all three are attached by
screws on each corner. The sample holder is divided in two parts (i.e., top and bottom, Fig.
S1c) with a Teflon spacer between them. Aerosol samples collected on the glass coverslips
(Fig. 2a) are placed between the two parts of the sample holder. Afterwards, the two parts of
the sample holder are fixed in place with four screws. As shown in Figure S1b, a circular
glass window is placed on the top of the sample holder to avoid any interference of the
ambient air. Finally, an optical microscope (Axiolab Zeiss, Germany) with a 5x/0.12
magnification objective is coupled to the sample holder (Fig. 3).


To maintain the temperature of the cooling block at ca. -80°C, polydimethylsiloxane
circulates through it with the help of a pump inside the cooling bath (PRO-RP1090,
LAUDA). A humid/dry air system is required to form liquid droplets on the aerosol particles
deposited on the glass coverslips. While humid air is obtained when the nitrogen (grade 4.8,
INFRA) passes through a bubbler filled up with pure water (LAL Reagent Water, Associates
of Cape Cod, Inc.), dry air is generated when the nitrogen is conducted towards the sample
holder. The humid and dry air are directed towards the sample holder by stainless steel tubes
(Swagelok), and the flows are controlled by four Swagelok valves (Fig. 3). Each experiment
is recorder by a video camera (MC500-W, JVLAB) located on top of the microscope and the
video is used to for further analysis. The temperature at the center of the sample holder was





obtained with a resistance temperature detector (RTD) with a ± 0.1°C uncertainty. The RTD is connected to a Fieldlogger device (RS485, NOVUS) to acquire the data (Fig. 3).

Given that the aim of the UNAM-DFT is to mimic ice particle formation via immersion freezing, a full experiment is performed as follows. At room temperature, the coverslip containing the aerosol particles were placed with the cold stage. Afterwards, the sample holder is isolated from the ambient atmosphere by tightening the screws and by positioning the circular glass window on top of it. Then, the sample holder is placed on top of the heating block when the temperature is at 0°C (Fig. 2b). Humid air is directed towards the sample

holder to induce liquid droplet formation by water vapor condensation, as shown in Figure 2c. Once the droplets have reached the desired size, dry air is directed toward the sample holder to shrink the droplets to minimize the contact between them (Fig. 2d). Approximately 30-40 droplets are formed on each glass coverslip. Afterwards, the humid/dry air system is closed, and the sample holder completely isolated by closing the valves located on either

side. Finally, the temperature of the sample holder is decreased from 0°C to -40°C at a cooling rate of 10°C min$^{-1}$. During the temperature ramp the droplets freeze promoted by the aerosol particles immersed within the droplets (Fig. 2e); otherwise, they are expected to freeze homogeneously close to -38°C.

***Frozen fraction***

The temperatures from the sample holder and the videos are analyzed to determine the freezing temperature of each droplet. The first metric employed was the frozen fraction ($F_{ice}$), which was obtained with the following equation:

$$F_{ice} = \frac{N_{ice}}{(N_{ice} + N_{droplets})} \qquad (1)$$

where $N_{ice}$ is the number of frozen drops (dimensionless) and $N_{droplets}$ is the number of unfrozen droplets (dimensionless) (Kanji et al., 2017).

 ***INP concentration***

The INP number concentration was calculated using the following expression:





$$[INP(T)] = -\ln\left(\frac{N_u(T)}{N_0}\right) \cdot \left(\frac{A_{deposit}}{A_{DFT}V}\right) \cdot N_0 \cdot f_{ne} \cdot f_{nu,0.25-0.10mm} \quad (2)$$

where $N_u(T)$ is the number of unfrozen droplets (L$^{-1}$) at a temperature $T$ (°C), $N_0$ is the total number of droplets (dimensionless), $A_{deposit}$ is the total area of the aerosol particles deposit on the MOUDI hydrophobic glass cover slip (cm$^2$), $A_{DFT}$ is the area of the sample analyzed by the DFT (cm$^2$), $V$ is the volume of air sampled by the MOUDI (L), $f_{ne}$ is a correction factor to account for the uncertainty associated with the number of nucleation events in each experiment (dimensionless), and $f_{nu}$ is a correction factor to account for changes in particle concentration across each MOUDI sample (dimensionless). More details can be found in Mason et al., (2015a).

### 2.2.2. Aerosol number and mass concentration

The particle size distribution for diameters ranging between 0.3 to 25 μm was obtained with optical particle counters (LasAir II 310A and LasAir III 310C, PMS, see Table 1), operated at 28.3 L min$^{-1}$ and at a sampling rate of 11 Hz. Also, the particle mass concentration, i.e., PM$_{10}$ and PM$_{2.5}$ were obtained with a FH62C14 Thermo Scientific Inc., operated at a flow rate of 16.7 L min$^{-1}$.

### 2.2.3. Chemical Composition

For the chemical analysis, aerosol particles were collected on 47 mm Teflon filters (Pall Science) for 48 h in Merida 2017, Merida 2018, and Sisal 2017 and for 24 h in Sisal 2018. The aerosol particles were collected using a Mini-Vol (TAS, Airmetrics), a Partisol (2025i, Thermo Fisher Scientific), and a cascade impactor (MOUDI 100NR, MSP) (see Table 1).

The Teflon filters were analyzed by X-ray fluorescence (XRF) to determine the elemental composition, using the X-ray spectrometer at the Laboratorio de Aerosoles, Instituto de Fisica, UNAM (Espinosa et al., 2012). Oxford Instruments (Scotts Valley, CA, USA) X-ray tube with an Rh anode and an Amptek X-123SDD spectrometer (Bedford, MA, USA) were used. The tube was operated at 50 kV and a current of 500 μA, irradiating during 900 s per spectrum. A set of thin film standards (MicroMatter Co., Vancouver, Canada) was used for





the calibration of the instrument. The spectra were analyzed using the AXIL (QXAS)
software.

### 2.2.4. Meteorological Variables

Wind speed, wind direction, air temperature, and RH were obtained using meteorological
stations. During Sisal 2017, a Davis (VANTAGE PRO2) meteorological station localized 20
m away from the other instruments was used. Meteorological data during the four other
sampling periods were obtained from meteorological sensors from the University Network
of Atmospheric Observatories (RUOA), installed at SC-UADY and EI-UNAM.
Additionally, back trajectories of the air masses arriving in Merida and Sisal were estimated
with the hybrid single-particle lagrangian integrated trajectory (HYSPLIT) model from the
National Oceanic and Atmospheric Administration (NOAA) for 72 hours and 13 days.

### 3. Results and discussion
### 3.1. UNAM-DFT performance

In order to evaluate the behavior of the UNAM-DFT, blank experiments with pure water
(LAL Reagent Water, Associates of Cape Cod, Inc.) were performed. In this case, new glass
coverslips were placed on the sample holder, and droplets (d= 45-250 µm) were formed on
them by condensation of water vapor on the substrate surface. The freezing temperature of
each droplet was later determined.


Figure 4 shows that most of the droplets (i.e., 95%) froze at temperatures below -36°C (rose
shaded area). The freezing temperatures of the blank experiments are in agreement with
previous homogeneous freezing values reported for liquid droplets using different methods
(Welti et al., 2012; Kohn et al., 2016; Nagare et al., 2016). Therefore, this indicates that the
substrate does not having a large impact in heterogeneous freezing results shown below. The
small differences between the present and literature homogeneous freezing curves can be
attributed to differences in the droplets size, the purity of the water, the exposure time of the
droplets to low temperatures, and the technique. For example, Kohn et al. (2016) used the
Portable Immersion Mode Cooling chAmber (PIMCA) coupled to the Portable Ice



Nucleation Chamber (PINC) with droplet sizes ranging from 10 µm to 14 µm in diameter (dashed blue line). Welti et al. (2012) (red stars) and Nagare et al. (2016) (magenta asterisks) used the Immersion Mode Cooling chAmber - Zurich Ice Nucleation Chamber (IMCA-ZINC), with ~20 µm droplets in diameter. Note that droplet size obtained with the UNAM-DFT are much larger than those used in the aforementioned studies. Additionally, the

homogeneous freezing experiments in the UNAM-DFT take around 4 min, in contrast with the residence time of droplets in the PIMCA-PINC and the IMCA-ZINC which are less than 10 s. Iannone et al. (2011) and Wheeler et al. (2015) also performed homogeneous freezing experiments using a DFT. Both studies found that 90% of the droplets froze at -37°C (brown diamonds and cyan triangles, respectively). Although the total time of the experiments

between the present study and those performed by Iannone et al. (2011) and Wheeler et al. (2015) are comparable, the size of the droplet may be slightly different.

Additionally, as shown by Lacher et al. (2020), INP concentrations reported by the UNAM-DFT during the Puy de Dôme ICe Nucleating Intercomparison Campaign (PICNIC) are in

good agreement with the values reported by other online and offline techniques such as the Portable Ice Nucleation Experiment Instrument (PINE), the Ice Nucleation Spectrometer of the Karlsruhe of Technology (INSEKT), the FRankfurt Ice nucleation Deposition freezinG Experiment (FRIDGE), the Colorado State University Ice Spectrometer (IS), the Ice Nucleation Droplet Array (INDA), the Leipzig Ice Nucleation Array (LINA), and the LED

based Ice Nucleation Detection Apparatus (LINDA). The results in Lacher et al. (2020) indicate that this newly built system is very robust, with a high level of confidence.

### 3.2. Chemical composition vs. air mass

The elemental chemical composition obtained by XRF for the three different air masses, MA,

BB, and AD, is shown in Figure 5. The pie charts shown into the blue, green, and orange squares are the results obtained for the MA, BB, and AD, respectively. The results of 29th January, 02nd April, and 06th July correspond to the background composition for each period. Important differences in elemental composition are evident depending on the source of the air mass.






Elemental composition of samples obtained in Sisal (Figure 5a) indicate that chlorine, sodium, sulfur, calcium, potassium, and magnesium were the most abundant always influenced by MA. The presence of these elements agrees with elemental composition reported in studies focused on MA (Parungo et al., 1986; Xia and Gao, 2010; Prather et al.,

2013). Prather et al. (2013) classified MA particles in four categories: inorganic sea salts (SS), mainly composed by NaCl; inorganic sea salt plus organic carbon (SS-OC); biological particles; and organic particles (OC) where organic species can be coupled with ions such as sulfur, magnesium, and calcium. O'Dowd et al. (2004), Ovadnevaite et al. (2014), and Lee et al. (2015) reported that the fraction of SS in the marine aerosol was 74%, 14-85%, and

72%, respectively. Furthermore, the high percentage of sulfur is associated with dimethyl sulfur emissions related to the marine biological activity linked to phytoplankton blooms (Yoch, 2002; Barnes et al., 2006). Herrera et al. (1996) characterized the phytoplankton in the GoM between 1999 and 2002, finding that diatoms, dinoflagellates, and chlorophytes are the most abundant type of phytoplankton of the coast of Sisal. Therefore, these

microorganisms could contribute to the sulfur emissions in this zone.

Sulfur is the most abundant element observed in Merida under background conditions (Figure 5b, left panel). The presence of sulfur can be attributed to anthropogenic sources, such as light- and heavy-duty vehicles with diesel engines that are prevalent in Merida. Sodium is

also observed in high percentage and can be attributed to MA advected into the city. Sulfur and to some extent also sodium, remain as the most abundant elements under the influence of air masses associated with BB, but other elements increase their abundance. Reid et al. (2004) reported that particles emitted during BB could contain trace inorganic species (e.g., sulfates, nitrate, chlorine, calcium, and potassium) which are also present in Figure 5b.

Potassium is a key element emitted during BB; it is an important nutrient for the plants, which is absorbed in the woody material through their roots, and it is transported to all growing areas (Ackerman and Cicek, 2017). Moreover, the mass concentration of $PM_{2.5}$ positively correlated with K (> 0.6) (Fig. S2a), corroborating that the sampled air masses during this season contained particles emitted from BB. Note that Li et al. (2003) found that soot

aggregates emitted by BB may contain potassium salts ($K_2SO_4$, $KNO_3$, and KCl) and in minor



quantity, elements such as iron, calcium, and magnesium, all of which are also present in the Merida samples during the influence of BB.

Figure 5c shows the significantly higher abundance of elements such as silicon, aluminum,
iron, calcium, and magnesium, which are characteristic of mineral dust particles, during the AD period (Fig. 5c, middle and right panel) compared with the background composition (Fig. 5c, left panel). Additionally, positive correlation coefficients ($\geq 0.8$) between the mass concentration of $PM_{2.5}$ and Mg, Al, Si, K, Ti, and Fe were found (Fig. S2b). Rosinski et al. (1988) reported high percentages of these elements in aerosol particles collected during July
and August in 1986 over the GoM. The authors attributed this composition to air masses originated in the East (probably from Africa). Al, Si, Ca, and Fe are typically found in mineral dust particles (Linke et al., 2006; Querol et al., 2019). Note that Al, Si, Mg, and Ca were also present in the background sample obtained on 6th July (Fig. 5c, left panel). This is attributed to the karstic soil typical of the region, in which variable percentages of Ca, Mg, and S are
present.

Finally, figure S2c shows the total organic carbon concentration (TOC) for the BB and AD periods. This provides further evidence that the two air masses were completely different, with higher TOC concentrations (by 2.0 μg m$^{-3}$) observed during the BB period.


### 3.3. Aerosol size distribution vs. air mass

Figure 6 shows the average aerosol size distribution sampled under the influence of the different air masses: MA (cyan), BB (red), and AD (yellow). For all three air masses, the highest particle concentration was observed for particles with sizes between 0.3 μm and 0.5
μm. For particles larger than 0.5 μm the concentration was found to decrease with size. Out of the three air masses, AD reported the highest particle concentration for particles ranging from 0.3μm to 5.0 μm. Likewise, the highest particle concentration for particles between 5.0 μm and 25 μm was found in the MA air masses. Except for the smallest size bin (i.e., 0.3-0.5 μm), aerosol particles measured during the BB season, showed the lowest concentration in
comparison to the other two air masses.





The size distribution for MA measured in Sisal is comparable to those reported by Si et al. (2018) for three different marine sites, with lower concentrations (North Pacific, North Atlantic, and Arctic oceans). Generally, the coast of Sisal is very calm with moderate wave

activity; however, during the MA sampling period, two cold fronts hit the Yucatan Peninsula, increasing the wind speed up to 28 km h$^{-1}$. High wind speeds are known to cause sea spray aerosol emissions by bubble bursting. Figure S3a shows examples of back trajectories obtained with HYSPLIT during the presence of one of the cold fronts, indicating that air masses crossed the GoM prior to their arrival in Sisal.


The AD particle number concentration measured in Sisal was found to be lower compared with literature data. For example, Gong et al. (2020b) reported that the aerosol concentration measured in São Vicente island, Cape Verde, under the influence of AD for particles larger than 1.0 µm varied between 3 cm$^{-3}$ and 71 cm$^{-3}$. Kaaden et al. (2009) found that under the

influence of Saharan dust particles in Morocco, the mean aerosol concentration for particles larger than 1.0 µm was 70 cm$^{-3}$. The lower particle number concentration measured in the Yucatan Peninsula may be attributed to the long distance the AD particles travelled before arriving to Mexico. While crossing the Atlantic many particles can be removed from the atmosphere by dry deposition as clouds are typically absent.


Although coarse particles were measured during the BB season at a much lower concentration, those particles may likely be associated with background particles such as re-suspended dust. Field studies of BB emissions have mostly focused on submicron particles (e.g., Pósfai et al. 2003; Zhang et al., 2011). Guyon et al. (2005) found that during BB events

in the Amazonia, the maximum submicron particle concentration was around 100 cm$^{-3}$ in fresh and detrained smoke. On the other hand, Hungershoefer et al. (2008) found that the maximum accumulation and coarse mode aerosol concentrations were 1.0 cm$^{-3}$ and 10$^{-4}$ cm$^{-3}$, respectively, when burning different African biomass.

### 3.4. INP Concentration vs. air mass

The INP concentration (L$^{-1}$) as a function of temperature and air mass type for particles between 0.32 µm and 10 µm is shown in Figure 7a. Samples obtained under the influence of





MA cause freezing at the warmest temperatures of the three different air masses, with onset
freezing temperatures as high as -5°C. AD was able to catalyze ice particle formation at
temperatures below -9°C. BB particles were the least efficient with onset freezing
temperatures below -19°C. The onset freezing temperatures of the MA, BB, AD particles
collected in the Yucatan Peninsula are in very good agreement with the values reported by
Kanji et al. (2017).

The INP concentration for MA (cyan circles) was found to range from 0.068 $L^{-1}$ to 18.90 $L^{-1}$ at temperatures between -5°C to -31°C. The present results were found to be higher than
those reported by Mason et al., (2016) in the west coast of Canada when the air masses were
coming from the northwest (i.e., 0.10 $L^{-1}$ to 6.1 $L^{-1}$ at temperatures from -15 to -25°C), by Si
et al. (2018) and Irish et al. (2019) in the Arctic (0.004 $L^{-1}$ to 0.67 $L^{-1}$ at temperature ranging
from -15°C to -25°C), by Mason et al. (2016) in Alert (0.05 $L^{-1}$ to 0.99 $L^{-1}$, at temperatures
from -15°C to -25°C), and by DeMott et al. (2016) in Puerto Rico (0.0002 $L^{-1}$ to 0.02 $L^{-1}$ at
temperatures from -6°C to -24°C). However, the present results are similar to those reported
by Mason et al., (2016) in the east coast of Canada for southeastern air masses (0.38 $L^{-1}$ to
2.8 $L^{-1}$, at temperatures from -15°C to -25°C). Recall that the MA samples in Sisal were
collected during the winter and in the presence of cold fronts. Therefore, the high wind speeds
associated with the cold fronts could have played an important role in the high INP
concentrations measured in Sisal, as discussed in Ladino et al. (2019).

The aerosol particles collected during the AD period reported the highest INP concentrations
with values ranging between 0.071 $L^{-1}$ and 36.07 $L^{-1}$ at temperature from -9°C to -29°C
(yellow squares). As shown in Figure 5, the AD particles were enriched in Al, Fe, and Si,
which are present in different minerals and clays (Querol et al. 2019). Although the 2018 AD
season was stronger than the 2017 season, as characterized by higher $PM_{2.5}$ and $PM_{10}$
concentrations (Ramírez-Romero et al. 2020), the INP concentrations measured in 2018 in
the Yucatan Peninsula were lower than values reported elsewhere. For example, Ardon-
Dryer and Levin (2014) found that the INP concentration in the eastern Mediterranean ranged
between 0.16 $L^{-1}$ and 234 $L^{-1}$ at temperatures between -11.8°C and -28.9°C, and higher
values (up to $10^3$ $L^{-1}$) at temperature from -18°C to -29°C were reported by Reicher et al.





(2019). Price et al. (2018) found that in Cape Verde, the INP concentrations were one order

of magnitude higher (0.1 L$^{-1}$ and 10$^2$ L$^{-1}$, at temperatures from about -10°C to -25°C) than those measured in Sisal and Merida. Note that the measurements reported by Ardon-Dryer and Levin (2014), Price et al. (2018), and Reicher et al. (2019) were performed relatively close to the mineral dust source, i.e., the Sahara and Arabian deserts. The lower values measured in the Yucatan Peninsula during the AD period can be associated with the long

distance (> 8,000 km), that the mineral dust particles travelled before reaching Mexico. During the long-range transport, the physicochemical properties of the AD particles may experience chemical ageing, likely impacting their ice nucleating abilities, consistent with the results of Boose et al. (2016). Also, those particles may have been exposed to different dilution processes during long-range transport, likely reducing the concentration of INP.


The measured INP concentration during the BB period was found to range between 0.063 L$^{-1}$ and 10.21 L$^{-1}$ at temperatures from -19°C to -28°C (red diamonds), the lowest INP concentrations, on average, of the three aerosol types. BB particles have been reported as inefficient INP via immersion freezing, with a higher potential to catalyze ice particles at

temperatures below -40°C via deposition nucleation (Kanji et al. 2017). The concentration of INP measured in Merida during the BB periods are lower than those reported by Prenni et al. (2009), Mccluskey et al. (2014), and Levin et al. (2016). The lower concentrations reported in the present study can be attributed to the long distance between the burning areas (likely southern Mexico and Central America) and the sampling site. Additionally, it has been

shown that the type of fuel/biomass can determine the ice-nucleating abilities of the BB emitted parted. Mccluskey et al. (2014) found clear differences in the ice nucleating abilities of different prescribed burns and wildfires.

Given that the UNAM-MOUDI-DFT can report the INP concentration as a function of the

aerosol particle size, the contributions of the submicron and supermicron particles to the total INP concentrations were assessed for the three aerosol types. Figure 7b-d shows that supermicron particles are the major contributor (> 72%) to the INP concentrations for all aerosol types. Note that this contribution is largest for the MA. These results are in agreement with previous studies (e.g., Mason et al., 2015a; Mason et al. 2016; Ladino et al. 2019; Gong





et al. 2020a). The present results and the aforementioned studies highlight the importance of supermicron particles in ice cloud formation, and therefore, they should not be excluded when conducting field measurements.

### 3.5. Surface active site density ($n_s$)

Figure 8 summarizes the $n_s$ values calculated for the three periods, i.e., MA, BB, and AD. The $n_s$ values were derived at -15°C, -20°C, -25°C, and -30°C using Equation 3:

$$n_s(T) = \frac{[INP_s(T)]}{S_{tot}}, \quad (3)$$

where *[INP(T)]* is the INP concentration at temperature *T* and $S_{tot}$ is the total surface area of all aerosol particles. See the supplementary information for more details on how $S_{tot}$ was

calculated.

As shown in Figure S4, the $n_s$ values reported in the literature for MA, BB, and AD covers several orders of magnitude. The $n_s$ obtained for the MA and BB particles overlap with the range of $n_s$ values reported in the literature. The MA and BB $n_s$ values were found to agree

with those reported by DeMott et al. (2016) and Umo et al. (2015), respectively. At -15°C and -20°C the $n_s$ values for the AD period were found to agree with the values reported by Gong et al. (2020). However, in comparison to the literature, lower $n_s$ values were found for the AD period at temperatures below -20°C.

Although the AD particles reported the highest INP concentrations between -9°C and -29°C (Figure 7), Figure 8 shows that the MA had the highest $n_s$ values in the Yucatan Peninsula (at least in winter), followed by AD at temperatures above -25°C. On the other hand, BB particles had lower $n_s$ values at -20°C and -15°C, but comparable values to the other air masses at -25°C and -30°C.


### 4.  Conclusions

The UNAM-MOUDI-DFT apparatus was satisfactorily built at the Universidad Nacional Autónoma de México. This new device is able to study mixed-phase cloud formation via immersion freezing and is capable to discriminate the INP concentrations as a function of the





aerosol particle size from 0.32 μm to 10 μm. The performance of the UNAM-MOUDI-DFT
was assessed by conducting homogeneous freezing experiments with pure water. Its
performance was also evaluated against other online and offline cloud chambers during the
Puy de Dôme ICe Nucleating Intercomparison Campaign (PICNIC), discussed in a separate
publication. The overall performance of the new device was found to be very good, providing

high quality results. The UNAM-MOUDI-DFT is expected to help the scientific community
understand the main sources of INP at tropical latitudes and to provide information urgently
needed for the development of new ice nucleation parameterizations that include aerosol
particles emitted in the tropics.

Aerosol particles sampled under the influence of three different air masses (marine aerosol,
biomass burning, and African dust) were collected between January 2017 and July 2018 in
Sisal and Merida, Yucatan (Mexico). The three different aerosol types were characterized by
their physicochemical properties (i.e., elemental composition, aerosol concentration, and
particle size). Furthermore, the samples were analyzed by the newly-developed UNAM-

MOUDI-DFT device, allowing the evaluation of the impact of different aerosol particles in
mixed-cloud formation in the Yucatan Peninsula. This is the first such comprehensive study
ever conducted in Mexico and also at tropical latitudes.

The three different air masses were found to contain INP with different ice nucleating

efficiencies. The cold fronts, responsible for advecting marine aerosol particles onto the
Yucatan Peninsula in winter, were found to be an important source of efficient INP. Out of
the three different aerosol types analyzed in the present study, marine aerosol particles were
identified as having the highest $n_s$ values and the highest onset freezing temperatures.
Therefore, this is the aerosol type with the highest potential to impact the local hydrological

cycle. Although marine aerosol particles are ubiquitous in the Yucatan Peninsula as it is
surrounded by the GoM and the Caribbean Sea, synoptic phenomena such as cold fronts and
tropical cyclones may enhance their concentration and likely their organic content. Based on
the present results, it is desirable to conduct similar measurements during the phytoplankton
bloom season, under the influence of a tropical cyclone, and in October-November when

there is much lower probability of occurrence of BB, AD, and cold fronts.





In the absence of cold fronts, AD may be an important driver in mixed-phase cloud formation, and hence, in precipitation development. Given that these particles are not locally emitted, it is very important to improve the current understanding of the long-range transport of AD

particles to Mexico, including their inter-annual variability.

Under the influence of all three air masses, supermicron particles were found to be the major contributor to the total INP concentrations. Given that it is very common to exclude supermicron particles when conducting field measurements due to technical reasons, the

present results highlight the importance of including the sampling of supermicron particles in future field measurements.

*Data availability*. Data are available upon request to the corresponding author.

*Author contributions*. FC, GBR, and LAL designed the field campaigns and the experiments. FC, CRR, DC, WG, MG, AKB, and LL built the UNAM-MOUDI-DFT. FC, CRR, HAO, DR, TA, BF, and LAL carried out the aerosol measurements. JM and HAO performed the chemical analyses. GBR, DB, DR, BF, JSK, JYH, and LAL installed the equipment and provided the infrastructure for the ADABBOY project. FC, CRR, and LAL wrote the paper,

with contributions from all coauthors.

*Competing interest.* The authors declare that they have no conflict of interest.

**Acknowledgements**

The authors thank Juan Carlos Pineda, Javier Juarez, and Aline Cruz, for their invaluable help. We also thank David S. Valdes from CINVESTAV Merida for sharing the Sisal meteorological data. Finally, we thank the National Oceanic and Atmospheric Administration (NOAA) for facilitating the use of the surface maps and the HYSPLIT. This study was financially supported by the Dirección General de Asuntos del Personal

Académico (DGAPA), the Consejo Nacional de Ciencia y Tecnología (Conacyt), and the





Universidad Autónoma de Yucatán through grants PAPIIT IA108417, FC-2164, and SISPROY-FQUI-2018-0003, respectively.

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





**Table 1.** Summary of the sampling periods, the used instrumentation, and the type of collected samples (for chemical composition and INP analysis) from Merida and Sisal during 985 2017 and 2018. MA, BB, and AD refers to marine aerosol, biomass burning, and African dust, respectively.

| Aerosol Type | Place | 2017 | 2018 |
|---|---|---|---|
| MA | Sisal | 21th Jan-2th Feb<br>CPC 3010<br>LasAir II 310A<br>MOUDI 100NR*<br>MOUDI 100R ** | *** |
| BB | Merida | 13th Apr-31th May<br>CPC<br>LasAir II 310A<br>FH62C14<br>Partisol 2025i *<br>MOUDI 100R ** | 26th Mar-08th Apr<br>CPC 3010<br>LasAir II 310A<br>FH62C14<br>Partisol 2025i *<br>MOUDI 100R ** |
| AD | Sisal | *** | 3th Jul -16th Jul<br>LasAir II 310A<br>MiniVol*<br>MOUDI 100R ** |
| AD | Merida | *** | 3th Jul -16th Jul<br>LasAir III 310C<br>MiniVol*<br>MOUDI 100R** |

\* samples collected for chemical analysis
\*\* samples collected for INP analysis
\*\*\*samples were not collected during this period

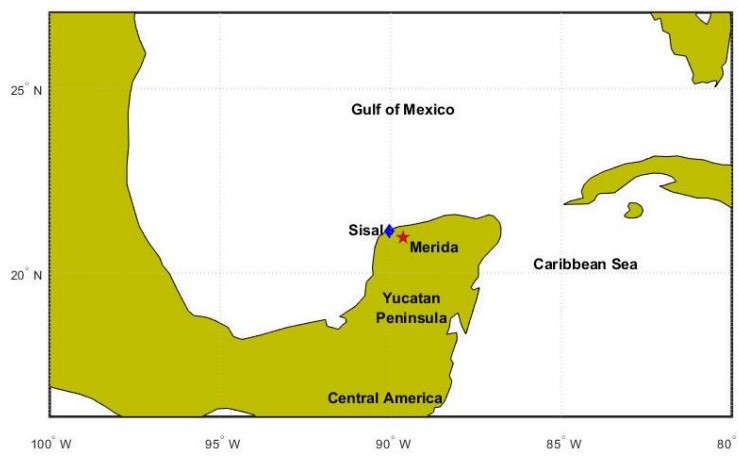


**Figure 1.** Map showing the Yucatan Peninsula and the sampling locations i.e., Sisal (blue diamond) and Merida (red star).





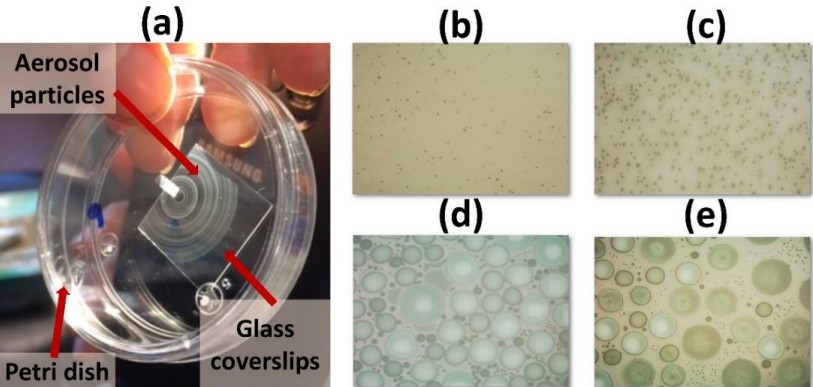

**Figure 2.** (a) aerosol particles collected on a hydrophobic glass coverslip, (b) aerosol
particles seen from the microscope (d=5.6-10 µm), (c) water vapor condensation on the
aerosol particles, (d) liquid droplets, and (e) frozen (opaque colors) and unfrozen droplets as
seen by the optical microscope.

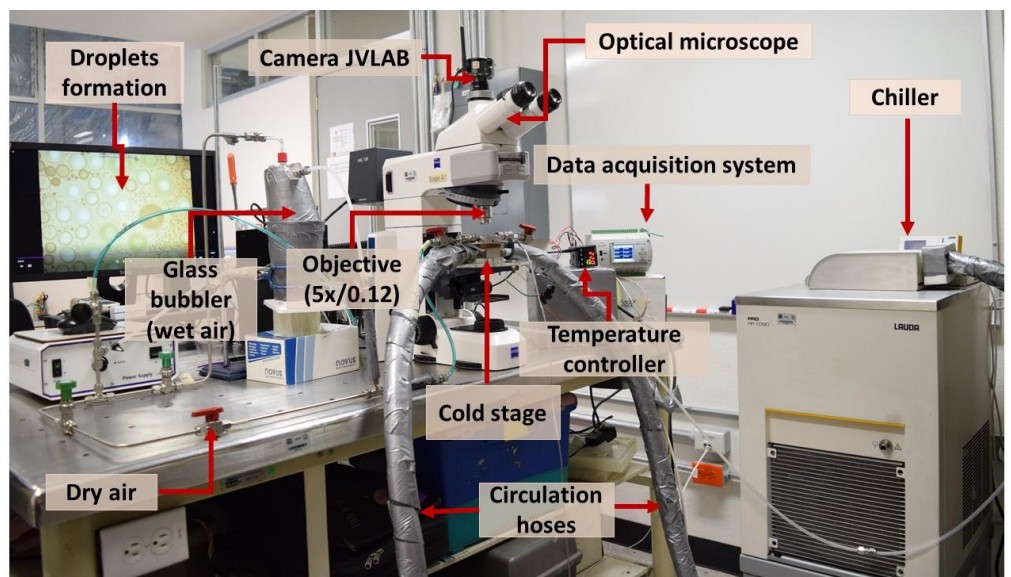


**Figure 3.** UNAM-DFT experimental setup with its main components. Figure S1 shows a
zoom of cold stage.


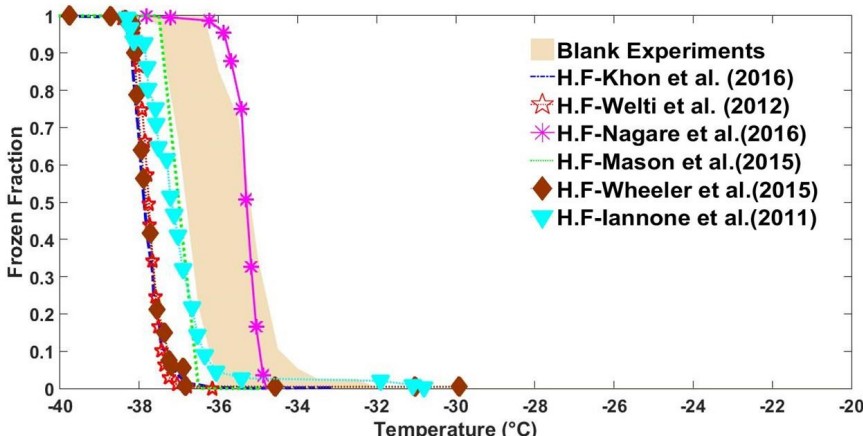

**Figure 4.** Activation curve for the blank experiments (rose shaded area) obtained with the UNAM-DFT using pure water. Colored lines correspond to homogeneous freezing (H.F.) curve literature data. Dashed blue line: Khon et al. (2016), Red stars: Welti et al. (2012), Magenta asterisks: Negare et al. (2016), Dotted green line: Mason et al. (2015), Brown diamonds: Wheeler et al. (2015), and Cyan triangles: Iannone et al. (2011).

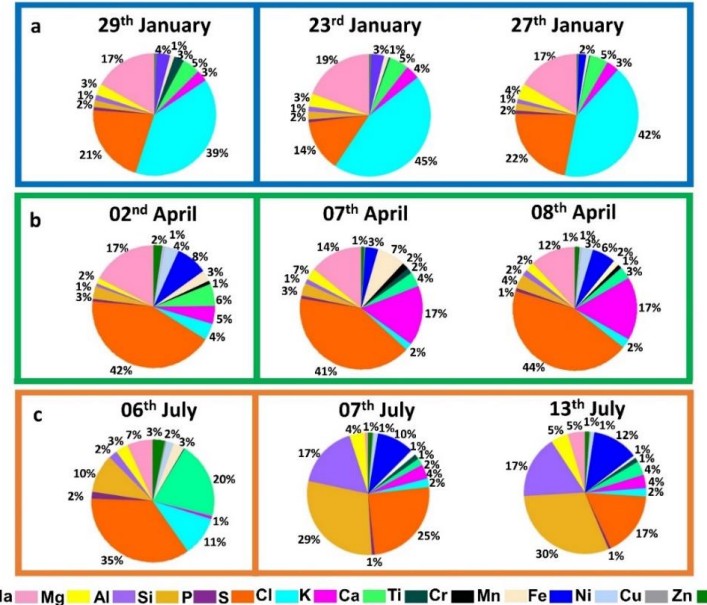


**Figure 5.** Elemental analysis obtained by x-ray fluorescence for (a) MA, (b) BB, (c) AD. Pie charts on the left correspond to background conditions (29th January, 02nd April and 06th July) while pie charts of the right correspond to days with atypical particle loadings.





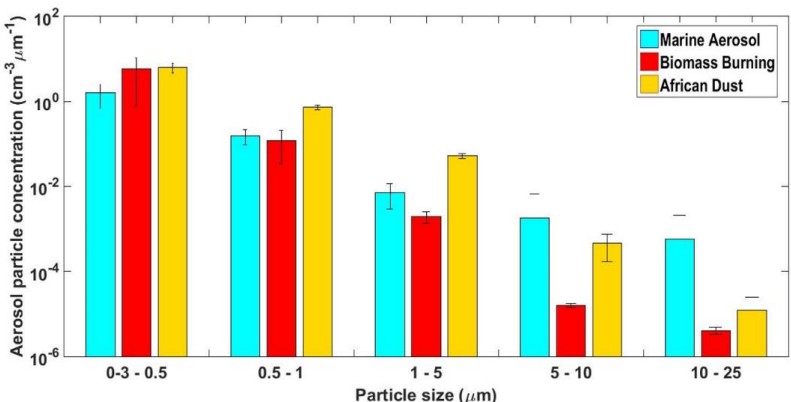

**Figure 6**. Average aerosol size distribution derived from the optical particle counters for the three air masses: MA (cyan bars), AD (yellow bars), and BB (red bars). The error bars represent the standard deviation.

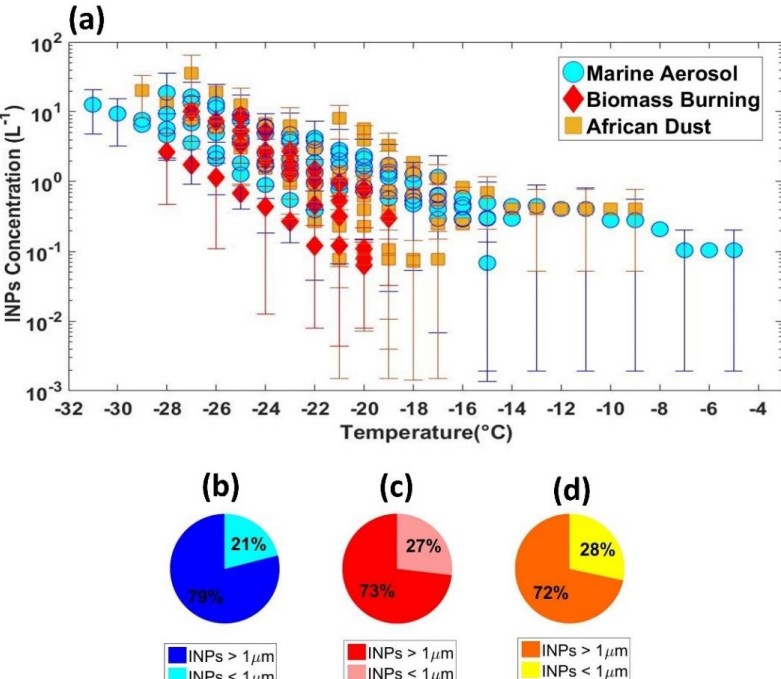

**Figure 7**. (a) INP concentrations as a function of temperature for MA (cyan circles), AD
(yellow squares), and BB (red diamonds) for aerosol particles with sizes ranging between
0.32 μm and 10 μm. The pie charts illustrate the contribution of supermicron and submicron
particles to the total INP concentration for (b) MA, (c) BB, and (d) AD.





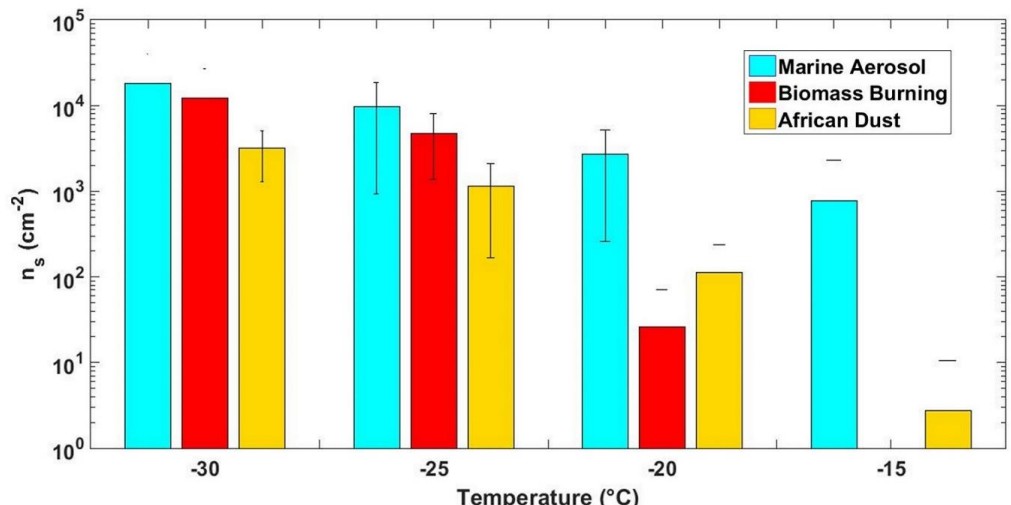

**Figure 8**. Surface active site density ($n_s$) as a function of temperature for MA, BB, and AD
particles. The error bars represent the standard deviation.