# Peer review of "Measurement report: Ice nucleating abilities of biomass burning, African dust, and sea spray aerosol particles over the Yucatan Peninsula"

_Atmospheric Chemistry and Physics, 2020_

## Referee Comment (RC1) · Anonymous Referee #1 · 22 Sep 2020

Review of "Measurement report: Ice nucleating abilities of biomass burning, African dust, and sea spray aerosol particles over the Yucatan Peninsula" by Córdoba et al.

**General comment** This study investigated ice-nucleating particles of size-segregated particles in tropical latitudes. Three types of common particles were studied: Marine, biomass burning and desert dust particles. Five field campaigns were conducted at two different sites in Mexico, and this is the first comprehensive study conducted in that area. The ice-nucleating abilities of the particles were tested in the new setup UNAM-MOUDI-DFT. INP concentrations varied between the different sampling events. The highest concentrations were observed during desert dust events, while the more ice-active particles were the marine aerosols, which sourced in the activity of cold front. Additionally, supermicron particles contributed most of the INPs in all particle types. This study is an important contribution to the cloud ice nucleation field, which presents a novel data collected in order to improve the understanding, characterization and quantification of ice nucleating particles. The sampling was well planned and the experiments were well designed. The results are well interpreted and clearly presented, and the data supports the conclusions. The manuscript is well written and within the scope of ACP. I recommend to accept this manuscript for publication in ACP after the authors will clarify and address the following comments.

**Major Comments**

1. The introduction section should include some crucial knowledge that discussed in the manuscript. What are the physical and chemical properties of an INP? The relation between particle size and ice nucleation is missing for example. The different chemistry of the three aerosol types which were studied and how this will affect their ice nucleation abilities, as well as how will atmospheric transport expected to affect their activity. Perhaps what type of minerals are commonly transported from North Africa and how will this change with particle size. It may be important to mention the presence of mineral phases which occurs in biomass burning aerosols and what are the main differences from desert dust.

2. In the section "The UNAM-DFT" there is no information about temperature calibration. I am concerned about the difference between the droplet's temperature and the detector temperature. Are they both located at the same surface? How accurate is the freezing point detection at 10 CPM ramp (L271)? What is the resolution of temperature acquired data? Is the temperature uncertainty (0.1 °C) remains constant down to -40 °C?

3. Consider to improve the English of the manuscript.

**Minor comments**

- 1. In the introduction section, starting from the third paragraph, there is a survey of atmospheric INP concentrations. The range of INP concentration depend on few factors, such as aerosol loading, sampling volume, and other technical issues, such as the droplets volume. It will be appreciated if the authors will add this important information to the introduction.
- 2. In **L304**, what was the reason for two different sampling periods? Is it the aerosol load? I think this should be explained in the text.

- 3. In L373, how 'background conditions' were determined in this study?
- 4. In **Figure 5**, how was the chemical composition quantified? It is written in the text that the elemental composition is obtained using XRF. There is a way to quantify the percentage of each element in this method? This should be mentioned in the text?
- 5. Figure 8, I do not understand what is the horizontal line over the bars at the higher temperatures.
- 6. Table S1, why these samples were chosen? Different air mass categories?

**Technical comments:**

L32: MicroOrifice should be Micro-Orifice

L33-35: Rephrase. Perhaps add "conditions" after "(AD)".

L36: Remove "characterizing".

L43: What do you mean in this final note?

**L72:** You referred to the different freezing mechanisms as "pathways" in L57, and as "modes" in L72. I suggest to use the same term along the manuscript.

L76: INP values in this paragraph are written in different formats. For example, 0.1 in L91

and  $2 \bullet 10^{-1}$  in L93. Perhaps best to use the same format.

L86: "eastern" should be "Eastern".

L120: Remove "the" before "sea".

L123: "eastern" should be "Eastern".

L130: What do you mean "similar than"? Rephrase.

L134: -35 deg. Celsius and not per Liter.

L138: Why suddenly here you place the temperature range in brackets?

L146-148: Rephrase, this sentence does not read well.

L163: Rephrase, this part of the sentence does not read well.

L166: Consider to add "the word "these" before "three".

L169: Remove "and methods"

L191: Merida is not affected by cold front as Sisal?

L196: Rephrase.

L202: Replace "efficiency" with "properties" or similar.

L210: Remove "used" or add :"in use" after "The MOUDI".

L213: Replace "time" with "period".

L235: Where was the thermocouple placed?

L255: Remove "to"

L261: Replace "were' with "was".

L320: What do you mean by "...from the University Network...". It is where it was located?

**L376: "**Indicate" should be "indicates".

L377: Replace "always" with "elements".

L389: "of the coast" should be "in the coast".

**L536-540:** This should be placed in the methodology section.

**Table 1.** Which CPC model was used for 2017 BB sampling in Merida?

---

## Referee Comment (RC2) · Anonymous Referee #2 · 8 Oct 2020

In the here presented study, atmospheric aerosol at two different locations on the Yucatan peninsula (Mexico) was examined, with a focus on Ice Nucleating Particles (INP). Three different types of air masses were examined, and besides for INP properties, general aerosol properties and results from a chemical analysis were made in addition.

The work is adding to the valuable growing body of studies on atmospheric INP from a region where previously no measurements had been made. The instrumentation used for the INP measurements is a still quite novel one. So throughout the text a feeling arose that comparisons with literature were sought, maybe to confirm the measurement

method. However, and this is one of my main criticism, are not always useful in the way they were made (more on that below at "specific comments).

While the structure of the text is fine, it is occasionally difficult to follow when too many numbers were given in the text instead of discussing the general results. Again, more about this is given below.

My main concern is about the method used for determining INP concentrations. Likely, I assume, the issue is about the respective description of the method, which is not sufficient. And again, find more on that below at "specific comments".

Additionally, the discrimination of the samples into three different phases was not motivated in a satisfactory way. It was not clearly described if/how samples were chosen amongst a larger batch or if they were really only discriminated through the time when they were samples. It was particularly shown that during the three different phases, there were variations in the chemical composition. That questions the discrimination into different phases as they were done. More on that also below.

Having said all that, the data and the manuscript as such are still valuable. So overall, I am sure that this manuscript can very likely be changed such that it can be published in ACP. However, major revisions are needed.

Specific comments:

Before going through the text line by line, up front here are my comments / concerns about three separate topics as indicated in the above text. I apologize for repetitions:

(1) Comparisons you make in the manuscript with data from literature do not always seem useful. While some of that is referred to again below, here my main concerns: You measure at different locations than other cited studies, with different distances to sources, so concentrations necessarily have to be different. In this respect, comparing concentrations does not make sense. Onset-temperatures are not useful for comparisons, either, as they depend on the amount of material sampled and the measurement

method used. Surface site densities (n_s) are useful for comparison, but only if the surface area is related to the same aerosol particle "type": some studies only look at one specific type of INP as for example only a mineral dust (chamber studies) and will result in n_s for these dust particles, while other studies examine atmospheric particles and use the surface area of all particles and will then result in n_s for this type of air mass. This is not well discriminated in your study and below some specific parts of the text related to this issue are given.

(2) Concerning your method: There are orders of magnitude more particles than there are INP, as you describe quite correctly in the introduction. Particles can be so small that you will not see them with your microscope (Fig. 2b will only show the largest ones). One droplet will certainly have a multitude of particles in it (compare Fig. 2b with 2d). That alone explains why you obtain useable results by looking at a comparably small number of droplets. But then, which fraction of the surface (i.e., of the sampled particles and therewith INP) is not included in droplets (between the visible droplets, but also on parts of the glass slide that are not examined at all)? And how many droplets will have more than one INP in them? How much does that influence the results? This is certainly included in the factors that are used in equation (2), but this is so essential to your method, that you can not only refer the reader to another publication. This needs to be explained in the manuscript (main text or at least in the SI). Some specific issues regarding your method already here:

line 237-238: How stable is your temperature calibration? I somehow assumed you would use a paste that ensures good thermal conductivity between different metal parts of a set-up. Have you ever tried that?

line 266 and Fig. 2: How is the desired size decided? How about droplets containing more than one particle (which will have to be the case)? Fig. 2 looks as if droplets "eat up" other droplets. - How about one droplet including several INP?

Maybe that all is included in the corrections factors given in equation (2), but as this is

a VERY crucial issue, this needs to be discussed in here, instead of only referring to another paper.

(3) The evaluation of the data based on different air masses which are assumed to predominantly appear during certain times of the year has its weaknesses. There are some high temperature INP that are ice active above -19°C for samples classified as both MA and AD which are absent for BB. Other than that, all data are quite similar. The comparisons of data from other locations, in which more clearly only certain air masses were measured seems a bit arbitrary, particularly as the results in here are not traced back to peculiarities of different air masses. Instead, the argument is that INP concentrations for MA are higher than at other marine locations because there were cold fronts with high winds while INP concentrations for AD are lower due to the larger distance. That all might make sense, but it could also be that AD and MA are similar in concentration as the sources in both cases were the same! Also, for BB, concentrations are low - but what was the background concentration before that air mass entered the BB region in Yucatan? Instead of comparing concentrations, which necessarily become lower further away from sources, it would be beneficial if only parameters as n_s would be compared. But as said above and again also below, care has to be taken to only compare to studies which were based on the same determination of the surface area (using all particles and not only a sub-fraction).

line 49: "more than 50% of the precipitation is initiated via the ice phase". It is not as easy as this - in the work you cite here, it is shown that this fraction depends on latitude and on land versus ocean. This number (50 %) for a global fraction is not given in the cited study, and I would advise you not to give such a clear number if it is not scientifically sound. "Large fractions" or something like that seems more appropriate.

line 64-65: "Only 1 in 10^5 to 10^6 of the aerosol particles can act as INP at temperatures higher than -38°C" - I know that something like that is often said, and I didn't check the citation here, but it is difficult to give such a fixed number, at least if it is not related to a temperature. If you look at the range of INP concentrations you've reported

in the abstract, you'll see that this already covers almost 4 orders of magnitude. And your values do not cover the whole temperature range from -38°C to 0°C, so the total range is MUCH larger. This shows that values within (only) one order of magnitude for the whole temperature range > -38°C, as you indicate it here, do not make sense. Saying "roughly one in a million at -20°C" would be a much better way of putting it. That can be estimated based on atmospheric samples in e.g., Petters & Wright, 2015.

line 122: An even broader dataset on Arctic ambient INP was published in Wex et al. (2019), even pointing towards an annual cycle in INP concentrations across the Arctic.

line 126ff: This comment concerns all about the Welti et al. (2020)-ACPD-paper: It is a bit strange to give the results such detailed for a study that is still in discussion. Also, it is not clear why you give these values so detailed, anyway. If you want to compare with values from your own results, then give these details later. Here, in the introduction, these many numbers only make the text very difficult to read. I suggest to delete all these numbers from line 123 to line 138 and instead give only one or a few important statements from this study. Also a small remark: Is it generally know what "N-temperate zone" is? I guess this is an expression used in the here cited study, and if you want to use it, you have to define it, too.

line 143-146: DeMott et al. (2003) even showed that INP from the Sahara were found over Florida.

line 158-159: Do you know the study by Schnell & Vali (1976)? They already report a dependence of INP concentrations with climate zones, albeit only based on leaf litter - read it and then decide yourself if it fits in here.

Table 1 and related text: CPCs are mentioned in Table 1 but not in the text. If they were used for the here presented analysis, add a description to the text. If they were not used in this study, delete them from the table. Also: Which CPC was used in Merida in 2017? Please specify in the table! Also: Mini-Vol was an abbreviation used in the text, in the table it's MiniVol. Unify!

line 196 and line 369-370: This is one of my main concerns: How were the samples included in this study chosen? Randomly? Or for which other reasons? Or were all collected filters included? If not, Did the choice of a subset influence the conclusions drawn from the study? As I understand, you just attribute different air masses to different seasons? Is that justified? There is a study by Wex et al. (2016), in which within one month on Barbados different air masses were observed (clean marine, marine with new particle formation, and African Dust), all advected across the Atlantic. That was the case in November, and again in April. Similarly, a certain air mass likely cannot be expected to be connected to a certain season at your sampling locations. Therefore: It at least has to be mentioned how you defined/chose the different air masses, and how you checked if all air masses during a sampling time belonged to the type of air mass you expect during this season.

line 215-216: How much time passed between sampling and analysis? Give that number here. Have you ever tested if the INP concentration is influenced by storage at temperatures above 0°C? Most studies I am aware of freeze their samples prior to the analysis.

line 370ff: You show in Fig. 5 and the related text, that "background conditions" were different from other days during the three different air mass types (particularly pronounced, to my understanding, for AD). How was this variability treated when you then summarized the results to obtain one value representative for the whole air mass (as done for Fig. 6 and Fig. 8)? This is, again, linked to how you chose the samples that are presented here.

line 392: Of course, if there is also a marine influence (which is to be expected), then part of the sulfur will also some from the same sources as described for MA above. Reformulate this passage accordingly!

line 429: Am I right to assume that you calculated the particle concentration based on the measured size distributions? Please mention explicitly how concentrations were

obtained!

line 453-454: There are trade wind cumuli across the Atlantic and they can produce warm rain! Just for fun I opened "windy.com" while writing this, checked for clouds between Africa and Yucatan, and (as I expected and as you will always see) there sure were several locations with intense cloud fields. Please correct this statement! It is clear, though, that particles are transported in "SAL" (Saharan Air Layer), and yes, "trickling out" from that indeed happens. But wet removal likely occurs, too.

line 461-463: Was the study you cite here based on a lab-experiment? If yes, then the dilution of the air also will have played a role, and then it is not clear to me how representative this is for atmospheric concentrations, and this quote might not be very meaningful.

line 467: In Tab. S1, there are 7, 5, and 4+4=8 samples for MA, BB and AD, respectively. The number of data-sets shown in Fig. 7a seems lower. Is that so, and if yes, why?

line 471-473: "onset freezing temperatures" are not a good value for judging a data-set, as they will depend on the amount of air sampled and the instrumentation used for the evaluation. At least add this information, or better delete this sentence completely.

paragraphs starting at 475 and at 488: It is much better to give comparative data in a figure instead of giving a range of numbers in a text. Additionally, a concentration of INP always has to be given at a temperature. Giving a range of concentrations spread over a range of temperatures is not helpful, and comparing concentrations obtained at different temperatures or for different temperature ranges is not meaningful, either. I suggest that you make an additional figure with your data with separate panels for the three air masses in which you always add the literature data you want to compare to. By reading this, one cannot understand the relation, and with this new figure you can avoid text filled with numbers which is difficult to read. And, as you quite correctly say, you are far away from the source, compared to the other studies to which you

are comparing to, so that comparing a concentration is not the best thing to do, as a concentration HAS TO decrease further away from a source. Comparing intrinsic INP properties, such as n_s, makes more sense.

line 489: Actually, INP concentrations for MA and AD are surprisingly similar, both in the range they cover and in their lowest and highest values (besides for these few high temperature data points - but here it might be reflected that some samples collected more air volume than others), so your statement here does not reflect that correctly. I am not even sure if the discrimination between MA and AD makes sense? Couldn't there also be dust particles for some of the filters collected during MA? The one trajectory shown in the SI doesn't say much about this.

line 511: Why is the trajectory shown for BB in the SI so short? If African dust would have been sampled during BB, that could dominate what you see. Alternatively, it could explain that the data is quite low if BB would not have contributed much to INP and the air mass did not come from Africa. A longer trajectory can help the discussion.

line 519: "(likely southern Mexico and Central America)" again made me wonder how you defined periods which were included in BB.

equation 3 and related text: Was the total surface area based on the aerodynamic diameter at the sampling RH used, here? From the parameter-names you use in the main text and in the SI, it seems that this is not the case? Please explain all this in more detail, at least in the SI. If the aerodynamic diameter at the sampling RH was used to calculate n_s, then why? Typically, dry particle sizes are used. It is not described anywhere, why all these transformations are given in the SI for the calculation of surface active site density (n_s) (aerodynamic to geometric, dry to RH). This certainly has to do with your optical size measurement, but all of this needs to be made clearer.

line 541ff, related to Fig. S4: Umo and Grawe only looked at ash particles, so their n_s relates to these, only. You include the surface area of all other particles, too, therefore it is logical that you end up with lower values. This should be said somewhere explicitly

in order not to confuse the readers. Something similar is true for most of the studies you used here for comparison: They examined a separate substance. Even Price only used the surface area of the (massive) amounts of dust particles, if I am not mistaken. It therefore makes perfect sense that mainly Gong fits as all others did not determine the n_s of an air mass with a mix of particles in it, but the n_s of one particle type. Also: are these separate components (starting with montmorillonite) your own measurements? Or else where do they come from? Adjust the text in the main manuscript accordingly!

line 582-583: As said above, INP concentrations of MA and AD were very similar as presented in Fig. 7 (as said, do not overinterpret the onset-temperatures), while indeed n_s was different. But are the INP in MA and SD clearly different? It's fine if this can be shown, but after going through the reviews, please adjust the text here accordingly.

Technical comments:

line 39-40: Add the temperatures at which the INP concentrations given here are valid - without that, it cannot be judged if these are high or low.

line 61: Change "its" to "their".

line 78: "Augustin-Bauditz et al. (2008)": I checked, as the date struck me as quite early, but this is a publication from 2014. This study cited here also focusses on K-feldspar as being the main mineral that drives ice nucleation, so it could additionally fit to what is said in line 84. (Discovery of this slip caused that I randomly checked other citations as well - Welti et al. (2020) should be cited as given below, and in the SI, at least Grawe et al. (2016) is missing in the References. - There may be other issues, please check.)

line 91-92: The samples for this study were not taken on Cape Verde, but flying over the ocean around Cape Verde, with some flights closer to the African continent. Maybe say "... and -23°C for airborne samples collected under dusty conditions around Cape Verde, Africa , ...".

line 94-95: If you follow the above advise, then here "in the same region" has to become "on one of the Cape Verde islands."

line 120: Delete the second "the" (at "in the Arctic the").

line 163: Add "of the role" between "understanding" and "that".

line 190: Concerning "20°C and 34°C": What are these two different temperatures (different seasons? day and night?)? Why are there two values here while above it was only one value for Merida and the Yucatan peninsula?

line 191: For the RH, the "+" part here makes no sense, as that ends up well above 100%.

line 196: Exchange "for presented" with either "for presentation" or "to be presented".

line 228: Better use "consists of" instead of "is integrated by".

line 230 and line 232 and line 236: cmˆ3 (not only cm)!

line 235: Where is this thermocouple? Maybe explicitly hint to Fig. S1 here, too.

line 251: How is the air "generated"? Maybe better: "added to the nitrogen flow when it is conducted ...".

line 255: Do you know the temperature variation across the glass plate? Please add this information here.

line 261: Be careful with the time form you use - this whole paragraph is in present tense, besides for this "were", here. -> are.

line 261: Instead of "with" better "on" or "in".

line 296: Either " diameters ranging from ... to ..." or "diameters between . . . and . . .".

line 297: Give the size range for which these two LasAir-instruments measured, in the text or table.

line 325: Why do you give two different lengths for the trajectories? Different for different samples / locations/ . . . ? Explain this in the text.

line 402: Replace "areas" with "parts".

line 402: Add "is" prior to "positively".

line 404: Add "a comparatively high mass fraction of" between "contained" and "particles".

line 427: Fig. 6 shows the concentration, not the distribution -> correct.

line 487: At least mention briefly, what the explanation is, that is given for your observation in the here cited publication.

line 558: Concerning "mixed-phase cloud formation": this should better be "immersion freezing" - you do not mimic the formation of a cloud, that would be up to cloud chambers.

line 561-564: As that PICNIC-study is not a topic described in your manuscript (besides that you mentioned that this other study exists), this should not be part of these conclusions here. Sentence should be deleted.

line 576: Concerning "and also at tropical latitudes." - How about Gong et al. (2020), which you've cited a number of times?

Fig. 4: In the caption: "Khon" -> "Kohn".

Fig. 7b: Black writing in a dark blue field is quasi invisible - change color of the text to white.

SI, Fig. S3: Typically the sequence you used so far is "MA", "BB", "AD" (following the sequence with which you sampled), so it is a bit confusing that the latter two are swapped, here. Please exchange.

SI, line 89: "base" -> "based".

Literature:

Augustin-Bauditz, S., H. Wex, S. Kanter, M. Ebert, F. Stolz, A. Prager, D. Niedermeier, and F. Stratmann (2014), The immersion mode ice nucleation behavior of mineral dusts: A comparison of different pure and surface modified dusts Geophys. Res. Lett., 41(20), 7375-7382, doi:10.1002/2014GL061317.

DeMott, P., K. Sassen, M. R. Poellot, D. Baumgardner, D. C. Rogers, S. D. Brooks, A. J. Prenni, and S. M. Kreidenweis (2003), African dust aerosols as atmospheric ice nuclei, Geophys. Res. Lett., 30(14), 1732, doi:1710.1029/2003GL017410.

Grawe, S., S. Augustin-Bauditz, S. Hartmann, L. Hellner, J. B. C. Pettersson, A. Prager, F. Stratmann, and H. Wex (2016), The immersion freezing behavior of ash particles from wood and brown coal burning, Atmos. Chem. Phys., 16, 13911–13928, doi:10.5194/acp-16-13911-2016.

Petters, M. D., and T. P. Wright (2015), Revisiting ice nucleation from precipitation samples, Geophys. Res. Lett., 42(20), 8758-8766, doi:10.1002/2015gl065733.

Schnell, R. C., and G. Vali (1976), Biogenic ice nuclei: Part I. Terrestrial and marine sources, J. Atmos. Sci., 33, 1554-1564.

Welti, A., E. K. Bigg, P. J. DeMott, X. Gong, M. Hartmann, M. Harvey, S. Henning, P. Herenz, T. C. J. Hill, B. Hornblow, C. Leck, M. Löffler, C. S. McCluskey, A. M. Rauker, J. Schmale, C. Tatzelt, M. van Pinxteren, and F. Stratmann (2020), Ship-based measurements of ice nuclei concentrations over the Arctic, Atlantic, Pacific and Southern Ocean, Atmos. Chem. Phys. Discuss., 20, doi:10.5194/acp-2020-466.

Wex, H., K. Dieckmann, G. C. Roberts, T. Conrath, M. A. Izaguirre, S. Hartmann, P. Herenz, M. Schäfer, F. Ditas, T. Schmeissner, S. Henning, B. Wehner, H. Siebert, and F. Stratmann (2016), Aerosol arriving on the Caribbean island of Barbados: Physical properties and origin, Atmos. Chem. Phys., 16, 14107–14130, doi:10.5194/acp-16-14107-2016.

Wex, H., L. Huang, W. Zhang, H. Hung, R. Traversi, S. Becagli, R. J. Sheesley, C. E. Moffett, T. E. Barrett, R. Bossi, H. Skov, A. Hünerbein, J. Lubitz, M. Löffler, O. Linke, M. Hartmann, P. Herenz, and F. Stratmann (2019), Annual variability of ice nucleating particle concentrations at different Arctic locations, Atmos. Chem. Phys., 19, 5293–5311, doi:10.5194/acp-19-5293-2019.

---

## Author Comment (AC1) · 21 Nov 2020

We would like to thank the Reviewers for their constructive suggestions, which helped us to improve the manuscript. Specific answers and manuscript modifications related to the Reviewers comments are given below in red text.

**Reviewer 1**

**General comment.** This study investigated ice-nucleating particles of size-segregated particles in tropical latitudes. Three types of common particles were studied: Marine, biomass burning and desert dust particles. Five field campaigns were conducted at two different sites in Mexico, and this is the first comprehensive study conducted in that area. The ice-nucleating abilities of the particles were tested in the new setup UNAM-MOUDI-DFT. INP concentrations varied between the different sampling events. The highest concentrations were observed during desert dust events, while the more ice-active particles were the marine aerosols, which sourced in the activity of cold front. Additionally, supermicron particles contributed most of the INPs in all particle types. This study is an important contribution to the cloud ice nucleation field, which presents a novel data collected in order to improve the understanding, characterization and quantification of ice nucleating particles. The sampling was well planned and the experiments were well designed. The results are well interpreted and clearly presented, and the data supports the conclusions. The manuscript is well written and within the scope of ACP. I recommend to accept this manuscript for publication in ACP after the authors will clarify and address the following comments.

A/ We thank the reviewer for her/his careful and detailed review and for his/her positive evaluation of the manuscript.

**Major Comments**
1. The introduction section should include some crucial knowledge that discussed in the manuscript. What are the physical and chemical properties of an INP? The relation between particle size and ice nucleation is missing for example. The different chemistry of the three aerosol types which were studied and how this will affect their ice nucleation abilities, as well as how will atmospheric transport expected to affect their activity. Perhaps what type of minerals are commonly transported from North Africa and how will this change with particle size. It may be important to mention the presence of mineral phases which occurs in biomass burning aerosols and what are the main differences from desert dust.
A/ We thank the reviewer for the suggestion. To provide a better context to the discussion section, the following text was added to the revised manuscript.

Lines 74-79: "Although it is not completely clear which physicochemical properties make an aerosol particle a good INP, its size (diameter >0.5 μm), crystal structure (similar to the ice lattice structure), chemical composition (hygroscopicity and ability to form hydrogen bonds), insolubility (solid surface for the ice embryo formation), and the presence of active sites have been reported as key factors (Pruppacher and Klett, 1997; Murray et al., 2012; Kanji et al., 2017)"

Lines 94-97: "Mineral dust is mainly composed of clays (e.g., kaolinite, montmorillonite, illite), feldspars and quartz, among others (e.g., Zimmermann et al., 2008; Ardon-Dryer and Levin, 2014). The ice nucleating abilities of pure minerals, clays, and surrogates of natural mineral dust particles have been evaluated in the immersion freezing mode."

Lines 133-138: "It is important to note that aerosol particles emitted by BB contain smoke and ash (Kanji et al., 2017). In general, BB particles are enriched in organic compounds, and to a lesser extent trace metals (Saarikoski et al., 2007). Jahn et al. (2020) showed that BB particles may contain calcite, sylvite, dolomite, potassium sulfate, and quartz. However, the mineral composition will differ depending on the combustion conditions, biomass source, and the chemical composition, among others.

Lines 141-143: "It has been shown that the composition of the marine aerosol (MA) strongly depends on the particle size, with the highest organic content found in submicron particles (O'Dowd et al., 2004; Prather et al., 2013; Quinn et al., 2015)."

Lines 191-194: "During long-range travel, the physicochemical properties (e.g., size, composition, morphology, hygroscopicity, etc.) of the transported aerosol particles can be modified due to atmospheric aging, thus, likely affecting their ice nucleating abilities (e.g., Kanji et al. 2017)."

2. In the section "The UNAM-DFT" there is no information about temperature calibration. I am concerned about the difference between the droplet's temperature and the detector temperature. Are they both located at the same surface? How accurate is the freezing point detection at 10 CPM ramp (L271)? What is the resolution of temperature acquired data? Is the temperature uncertainty (0.1 °C) remains constant down to -40 °C?

A/ The RTD used to monitor the droplet's temperature is placed directly below the area we analyze with the DFT, as shown in Figure S1. This means that the temperature sensor is indeed on the same surface the droplets are located. Given that the temperature ramp is controlled by a thermocouple (also located on the same surface as the droplets and the RTD, Figure S1), a calibration of the readings reported by the RTD and the thermocouple against the temperature of the brand-new cooling bath was performed as shown in Figure A1.

As shown in Figure A1, the temperature data reported by the RTD is in good agreement with data reported from the Chiller which was calibrated by the manufacturer in Germany.

The precision of the temperature is 0.1°C and the uncertainty reported by the RTD varies between 0.1°C and 0.56°C when the temperature decreases. This is now acknowledged in the revised manuscript. Lines 294-296: "The temperature at the center of the sample holder was obtained with a resistance temperature detector

(RTD) with a precision of 0.1°C and an uncertainty of 0.1°C to 0.56°C between 0°C and -40°C, respectively."

[Figure]

Figure A1. Comparison of temperature scans derived for the thermostat LAUDA PRO RP 1090 vs a resistance temperature detector (RTD) and a thermocouple immersed into the cooling bath. Each line represents the average of two experiments with its standard deviation.

3. Consider to improve the English of the manuscript.

A/ The text has been carefully revised by one the co-authors (D. Baumgardner), a native English speaker.

**Minor comments**

1. In the introduction section, starting from the third paragraph, there is a survey of atmospheric INP concentrations. The range of INP concentration depends on few factors, such as aerosol loading, sampling volume, and other technical issues, such as the droplets volume. It will be appreciated if the authors will add this important information to the introduction.
A/ Thank you for the suggestion. The following text was added to the revised manuscript.

Lines 89-92 "Note that a direct comparison of the INP concentrations reported in different studies needs to be considered carefully, as there are factors associated with each technique/method such as sampling volume, droplet size, aerosol loading, aerosol particle size range, temperature range, among others, that could be influencing specific results.

2. In **L304**, what was the reason for two different sampling periods? Is it the aerosol load? I think this should be explained in the text.
A/ The reviewer is correct, the main reason was the aerosol loading. This is acknowledged in the revised manuscript. Lines 361-362: "Note that there were two

different sampling periods in Sisal to evaluate the interannual variability in aerosol loading".

3. In L373, how 'background conditions' were determined in this study?
A/ The background was defined as the lowest $PM_{2.5}$ and $PM_{10}$ values within each sampling period when the chemical composition was available. This is acknowledged in the revised manuscript.

Lines 437-439: "The background days were defined as those with the lowest $PM_{2.5}$ and $PM_{10}$ values within each sampling period when the chemical composition was available."

4. In **Figure 5**, how was the chemical composition quantified? It is written in the text that the elemental composition is obtained using XRF. There is a way to quantify the percentage of each element in this method? This should be mentioned in the text?
A/ The chemical composition was quantified using the following equation (Espinosa et al., 2010):

$$C_i = \frac{N_{X,i} A_F}{k_i QV}$$

Where $C_i$ is the concentration ($\mu g\ m^{-3}$) of element i, $N_{X,i}$ is the number of photons in the peak of spectra obtained by XRF of element $i$. $A_F$ is the filter area used in the sampling, $k_i$ is the signal perceived by the detector, $Q$ is the total charge incident and $V$ is the volume of air of the sampling collection. Finally, this concentration was converted into ng $m^{-3}$.

The following text was added to the revised manuscript. Lines 373-375: "The chemical composition was quantified using the methodology reported by Espinosa et al. (2010). Once the concentration of each analyzed element was obtained, the fraction (in percentage) with respect to the total mass concentration was derived."

5. **Figure 8**, I do not understand what is the horizontal line over the bars at the higher temperatures.
A/ The horizontal lines correspond to the error bars; however, there was an issue with MatLab that was corrected in the revised version of the figure.

6. **Table S1**, why these samples were chosen? Different air mass categories?

A/ The Yucatan Peninsula is affected by cold fronts between November and March, by biomass burning emissions from Central America and south of Mexico between January and May, and by African dust between June and August. Therefore, different samples were collected during each of the above mentioned periods of interest. Although several samples were collected during each sampling period for ice nucleation analysis, Table S1 lists the samples where ice nucleation for the full size range was available (i.e., from 0.32 $\mu m$ to 10 $\mu m$).

The Table's caption was modified as follows: "Table S1. Summary of a subset of samples (those with the full size range available i.e., from 0.32 μm to 10 μm) taken from Merida and Sisal during 2017 and 2018 to analyze the results presented in this study. MA, BB, and AD refer to marine aerosol, biomass burning, and African dust, respectively. *two samples were collected at different times during the same day.

**Technical comments:**

**L32:** MicroOrifice should be Micro-Orifice
A/ Corrected.

**L33-35:** Rephrase. Perhaps add "conditions" after "(AD)".
A/ The word "intrusions" was added after "(AD)".

**L36:** Remove "characterizing".
A/ Removed.

**L43:** What do you mean in this final note?
A/ The sentence was deleted as it was redundant.

**L72:** You referred to the different freezing mechanisms as "pathways" in L57, and as "modes" in L72. I suggest to use the same term along the manuscript.
A/ We agree with the reviewer and we decided to stick to "modes" in the revised manuscript.

**L76:** INP values in this paragraph are written in different formats. For example, 0.1 in L91 and 2•10-1 in L93. Perhaps best to use the same format.
A/ Thank you for pointing this out. This was fixed in the revised manuscript.

**L86:** "eastern" should be "Eastern".
A/ Corrected.

**L120:** Remove "the" before "sea".
A/ Removed.

**L123:** "eastern" should be "Eastern".
A/ Corrected.

**L130:** What do you mean "similar than"? Rephrase.
A/ The sentence was deleted in the revised manuscript.

**L134:** -35 deg. Celsius and not per Liter.
A/ We are sorry for the typo. This sentence was deleted in the revised manuscript.

**L138:** Why suddenly here you place the temperature range in brackets?
A/ The sentence was deleted in the revised manuscript.

**L146-148:** Rephrase, this sentence does not read well.
A/ The following text was modified. Lines 180-183: "The highest probability for mineral dust to reach the Caribbean and Mexico occurs during the mid-summer drought (MSD), a period of a relative minimum in precipitation within the rainy season, typically between mid-July and mid-August.

**L163:** Rephrase, this part of the sentence does not read well.
A/ The sentence was modified as follows. Lines 200-201: "there is an urgent need to improve the current understanding of the role of aerosol particles, emitted in tropical latitudes, in cloud formation.

**L166:** Consider to add "the word "these" before "three".
A/ Added.

**L169:** Remove "and methods"
A/ Removed.

**L191:** Merida is not affected by cold front as Sisal?
A/ Merida is also affected by cold fronts; however, the marine aerosol sampling was performed in Sisal, a site 50 m away from the shoreline with  anthropogenic influences.

**L196:** Rephrase.

A/ The text was modified as follows. Lines 235-237: "A subset of the samples (those with the full size range available i.e., from 0.32 um to 10 um) collected during the five campaigns was chosen for this study, as summarized in Table S1.

**L202:** Replace "efficiency" with "properties" or similar.
A/ "efficiency" was replaced by "abilities".

**L210:** Remove "used" or add :"in use" after "The MOUDI".
A/ Removed.

**L213**: Replace "time" with "period".
A/ Replaced.

**L235:** Where was the thermocouple placed?
A/ The thermocouple is placed in the holder sampling, next to the RTD. A tag was added to Figure S1 to indicate the position of the thermocouple.

**L255:** Remove "to"
A/ Removed.

**L261:** Replace "were' with "was".
A/ Replaced.

**L320:** What do you mean by "..from the University Network…". It is where it was located?
A/ The University Network of Atmospheric Observatories (RUOA, in Spanish is Red Universitaria de Observatorios Atmosféricos) is a set of meteorological and air pollution observatories located in different cities in Mexico such as Merida and Sisal.

**L376: "**Indicate" should be "indicates".
A/ ixed.

**L377:** Replace "always" with "elements".
A/ Replaced.

**L389:** "of the coast" should be "in the coast".
A/ Fixed.

**L536-540:** This should be placed in the methodology section.
A/ Thank you for the suggestion. This text was moved to Lines 343-348.

**Table 1.** Which CPC model was used for 2017 BB sampling in Merida?
A/ Thank you for pointing this out. However, the CPC was removed from Table 1 as it was not used at all in the present study.

**Reviewer 2**

In the here presented study, atmospheric aerosol at two different locations on the Yucatan peninsula (Mexico) was examined, with a focus on Ice Nucleating Particles (INP). Three different types of air masses were examined, and besides for INP properties, general aerosol properties and results from a chemical analysis were made in addition.

The work is adding to the valuable growing body of studies on atmospheric INP from a region where previously no measurements had been made. The instrumentation used for the INP measurements is a still quite novel one. So throughout the text a feeling arose that comparisons with literature were sought, maybe to confirm the measurement method. However, and this is one of my main criticism, are not always useful in the way they were made (more on that below at "specific comments).

While the structure of the text is fine, it is occasionally difficult to follow when too many numbers were given in the text instead of discussing the general results. Again, more about this is given below.

My main concern is about the method used for determining INP concentrations. Likely, I assume, the issue is about the respective description of the method, which is not sufficient. And again, find more on that below at "specific comments".

Additionally, the discrimination of the samples into three different phases was not motivated in a satisfactory way. It was not clearly described if/how samples were chosen amongst a larger batch or if they were really only discriminated through the time when they were samples. It was particularly shown that during the three different phases, there were variations in the chemical composition. That questions the discrimination into different phases as they were done. More on that also below.

Having said all that, the data and the manuscript as such are still valuable. So overall, I am sure that this manuscript can very likely be changed such that it can be published in ACP. However, major revisions are needed.

A/ We thank the reviewer for her/his careful and detailed review and for his/her positive evaluation of the manuscript. Below we answer each of the above mentioned concerns.

Regarding the used method, we would like to mention that a direct comparisons have already been made between the technique used in the current study and other apparatus commonly used to measure INP. Therefore, the apparatus has already been validated reasonably well and the point of the current study is not to validate the instrument.

- DeMott, P. J., et al. (2017). "Comparative measurements of ambient atmospheric concentrations of ice nucleating particles using multiple immersion freezing methods and a continuous flow diffusion chamber." Atmospheric Chemistry and Physics **17**(18): 11227-11245.

- Mason, R. H., et al. (2015). "The micro-orifice uniform deposit impactor–droplet freezing technique (MOUDI-DFT) for measuring concentrations of ice nucleating particles as a function of size: improvements and initial validation." Atmospheric Measurement Techniques **8**(6): 2449-2462.

**Specific comments:**
Before going through the text line by line, up front here are my comments / concerns about three separate topics as indicated in the above text. I apologize for repetitions:

**(1)** Comparisons you make in the manuscript with data from literature do not always seem useful. While some of that is referred to again below, here my main concerns: You measure at different locations than other cited studies, with different distances to sources, so concentrations necessarily have to be different. In this respect, comparing concentrations does not make sense. Onset-temperatures are not useful for comparisons, either, as they depend on the amount of material sampled and the measurement method used. Surface site densities (n_s) are useful for comparison, but only if the surface area is related to the same aerosol particle "type": some studies only look at one specific type of INP as for example only a mineral dust (chamber studies) and will result in n_s for these dust particles, while other studies examine atmospheric particles and use the surface area of all particles and will then result in n_s for this type of air mass. This is not well discriminated in your study and below some specific parts of the text

related to this issue are given.

A/ Although the samples from the present study were indeed collected in different places than those reported in the literature (this is one of the novel parts), our intention is to evaluate how our results compare with respect to other studies carried out in similar environments and/or influenced by the same aerosol types. This is a very common practice in the ice nucleation community as the likelihood that two studies, on the same topic, are performed in the same place is extremely low. We are convinced that without these comparisons, we leave the readers without a clear context of where the present data stand out. Finally, the direct comparison of the INP concentrations and the $n_s$ values reported by different methods/techniques from samples collected in very diverse environments, altitudes, and seasons are commonly reported (e.g., Hoose and Mohler, 2012; Murray et al., 2012; Petters and Wright 2015; Kanji et al. 2017; Welti et al. 2020 etc.). The following text was added to the revised manuscript to acknowledge that such comparisons need to be treated carefully.

Lines 89-93: "Note that a direct comparison of the INP concentrations reported in different studies needs to be considered carefully, as there are factors associated with each technique/method such as sampling volume, droplet size, aerosol loading, aerosol particle size range, temperature range, among others, that could be influencing specific results.

(2) Concerning your method: There are orders of magnitude more particles than there are INP, as you describe quite correctly in the introduction. Particles can be so small that you will not see them with your microscope (Fig. 2b will only show the largest ones). One droplet will certainly have a multitude of particles in it (compare Fig. 2b with 2d). That alone explains why you obtain useable results by looking at a comparably small number of droplets. But then, which fraction of the surface (i.e., of the sampled particles and therewith INP) is not included in droplets (between the visible droplets, but also on parts of the glass slide that are not examined at all)? And how many droplets will have more than one INP in them? How much does that influence the results? This is certainly included in the factors that are used in equation (2), but this is so essential to your method, that you can not only refer the reader to another publication. This needs to be explained in the manuscript (main text or at least in the SI). Some specific issues regarding your method already here:
A/ We thank the reviewer for this suggestion. The paper is designed to focus more on the scientific data than on the method itself since the technique was already described in great detail by Mason et al. (2015). If we add the requested data it will have to go in the SI. However, we do not see the utility of repeating the information that it is already reported in Mason et al. (2015) in the SI.

line 237-238: How stable is your temperature calibration? I somehow assumed you would use a paste that ensures good thermal conductivity between different metal parts of a set-up. Have you ever tried that?
A/ The cold cell was built in order that the RTD cylinder perfectly fits into the designated orifice. Additionally, the RTD was secured with an elastic band. This ensures a good contact between the RTD and the cold cell. A single test using the

thermal conductivity paste described in Mason et al. (2015) was performed confirming the above conclusion.

**line 266** and Fig. 2: How is the desired size decided? How about droplets containing more than one particle (which will have to be the case)? Fig. 2 looks as if droplets "eat up" other droplets. - How about one droplet including several INP?
Maybe that all is included in the corrections factors given in equation (2), but as this is a VERY crucial issue, this needs to be discussed in here, instead of only referring to another paper.
A/ The desired droplet size is around 170 μm in diameter. Having smaller droplets will make it difficult to see the phase transition from liquid to ice as those droplets are quite dark. Similarly, having larger droplets reduces the number of droplets per experiment that resulted in low statistics. The following text was added to the revised manuscript. Lines 305-306: "Once the droplets have reached a diameter of ~170 μm (on average)".

Equation 2 considers the possibility of the existence of several particles within a droplet. Also, Equation 2 includes a correction of the analyzed area as the analyzed area with DFT is smaller (1.2 mm$^2$) compared with the total area of deposit of particles. The text was modified as follows.

Lines 335-341: "Equation 2 considers the possibility that multiple particles can be present within a droplet (Vali, 1971). As part of the data analysis, two corrections are included. The first is related to the total area covered by the particles deposited on the coverslips, which varies between 425 mm$^2$ and 605 mm$^2$ as a function of the MOUDI stage. The second is related to inhomogeneities in particle deposit. The correction factor associated with the nucleation events ($f_{ne}$) was calculated with a 95% confidence level (Mason et al., 2015a). More details on these corrections can be found in Mason et al., (2015a)."

**(3)** The evaluation of the data based on different air masses which are assumed to predominantly appear during certain times of the year has its weaknesses. There are some high temperature INP that are ice active above -19_C for samples classified as both MA and AD which are absent for BB. Other than that, all data are quite similar.
A/ The Yucatan Peninsula is affected by cold fronts between November and March, by biomass burning emissions from Central America and southern Mexico between January and May, and by African dust between June and August. Wind directions and synoptic patterns allow for the initial differentiation of air masses. And the different sampling campaign were intended to maximize the probabilities of encountering air masses with very distinct origins: marine air (MA), biomass burning (BB) and African dust (AD). Samples were collected during each of the periods of interest. The characteristics of each air mass were evaluated and assigned based upon the chemical composition of the aerosol particles collected during each period. As seen in Figure 5, aerosol composition is clearly different in each air mass category and in accordance to the literature (Figure 5). Also, the particle size distribution of

each sampling period was monitored as shown in Figure 6. Finally, Figure S3 shows that the air mass back trajectories for the MA, BB, and AD are very distinct.

Regarding the apparent similarity of the INP concentrations for all three periods, this is may be due to an optical effect. As shown below in Figure A2, when using different colors for each sample and period of interest, their differences are evident, given the broad range in values on the logarithmic y-axes.

[Figure]

Figure A2: INP concentration curves to each air mass using different colors per sample. (Top) MA, (medium) BB, and (bottom) AD.

The comparisons of data from other locations, in which more clearly only certain air masses were measured seems a bit arbitrary, particularly as the results in here are not traced back to peculiarities of different air masses. Instead, the argument is that INP concentrations for MA are higher than at other marine locations because there were cold fronts with high winds while INP concentrations for AD are lower due to the larger distance. That all might make sense, but it could also be that AD and MA are similar in concentration as the sources in both cases were the same! Also, for BB, concentrations are low - but what was the background concentration before that air mass entered the BB region in Yucatan? Instead of comparing concentrations, which necessarily become lower further away from sources, it would be beneficial if only parameters as n_s would be compared. But as said above and again also below, care has to be taken to only compare to studies which were based on the same determination of the surface area (using all particles and not only a sub-fraction).

A/ Please see our answer to Major point (1).

**line 49**: "more than 50% of the precipitation is initiated via the ice phase". It is not as easy as this - in the work you cite here, it is shown that this fraction depends on latitude and on land versus ocean. This number (50 %) for a global fraction is not given in the cited study, and I would advise you not to give such a clear number if it is not scientifically sound. "Large fractions" or something like that seems more appropriate.

A/ The text was modified as follows. Lines 54-57: "Globally, precipitation in the Intertropical Convergence Zone (ITCZ) and the continents is dominated by cold rain, and therefore, it is very important to fully understand the formation of ice in clouds (Mülmenstädt et al., 2015).

**line 64-65**: "Only 1 in 10^5 to 10^6 of the aerosol particles can act as INP at temperatures higher than -38_C" - I know that something like that is often said, and I didn't check the citation here, but it is difficult to give such a fixed number, at least if it is not related to a temperature. If you look at the range of INP concentrations you've reported in the abstract, you'll see that this already covers almost 4 orders of magnitude. And your values do not cover the whole temperature range from -38_C to 0_C, so the total range is MUCH larger. This shows that values within (only) one order of magnitude for the whole temperature range > -38_C, as you indicate it here, do not make sense. Saying "roughly one in a million at -20_C" would be a much better way of putting it. That can be estimated based on atmospheric samples in e.g., Petters & Wright, 2015.

A/ We agree with the reviewer that the number of aerosol particles acting as INPs is temperature dependent. However, Lines 64-65 (in the original manuscript) correspond to a general statement often reported in textbooks (e.g., Lohmann et al. 2016 page 226; Tomasi et al. 2017 pages 161 and 225) and ice nucleation papers (e.g., DeMott et al. 2010 and Boose et al. 2016, among others). Please note that our notation is exactly the same one used in Lohmann et al. (2016).

- DeMott, P. et al.: Predicting global atmospheric ice nuclei distributions and their impacts on climate, P. Natl. Acad. Sci., 107, 11 217–11 222, 2010.

- Boose, Y., et al.: Ice nucleating particles in the Saharan Air Layer, Atmos. Chem. Phys., 16, 9067-9087, https://doi.org/10.5194/acp-16-9067-2016, 2016.
- Lohmann, U., Lüönd, F., and Mahrt, F.: An Introduction to Clouds: From the Microscale to Climate, Cambridge University Press, Cambridge, https://doi.org/10.1017/CBO9781139087513, 2016.
- Tomasi, C., Fuzzi, S., y Kokhanovsky, A. (2017). Atmospheric aerosols: Life cycles and effects on air quality and climate. John Wiley & Sons.

**line 122**: An even broader dataset on Arctic ambient INP was published in Wex et al. (2019), even pointing towards an annual cycle in INP concentrations across the Arctic.

A/ Thank you for the suggestion. The reference was added.

- Wex, H. et al. (2019), Annual variability of ice nucleating particle concentrations at different Arctic locations, Atmos. Chem. Phys., 19, 5293–5311, doi:10.5194/acp-19-5293-2019.

**line 126ff:** This comment concerns all about the Welti et al. (2020)-ACPD-paper: It is a bit strange to give the results such detailed for a study that is still in discussion. Also, it is not clear why you give these values so detailed, anyway. If you want to compare with values from your own results, then give these details later. Here, in the introduction, these many numbers only make the text very difficult to read. I suggest to delete all these numbers from line 123 to line 138 and instead give only one or a few important statements from this study. Also a small remark: Is it generally know what "N-temperate zone" is? I guess this is an expression used in the here cited study, and if you want to use it, you have to define it, too.

A/ We understand the reviewer concern related to the Welti et al. (2020) study. However, it is important to note that the Welti et al. (2020) study is already accepted for its publication in ACP. We consider that it is very valuable to keep the Welti et al. (2020) study as it is a comprehensive study for marine aerosol, a key component of our manuscript.

Following the reviewer's suggestion, the text was modified as follows. Lines 153-159: "Welti et al. (2020) presented an analysis of the INP concentrations of ship-based measurements for different zones in the Arctic, Atlantic, Pacific, and the Southern Ocean. The different climatic zones covered in the Welti et al. (2020) study were divided depending on the latitude. For the tropical zone (23.5°N - 23.5°S), the authors report that the concentration of INPs vary between 0.001 $L^{-1}$ to 0.1 $L^{-1}$ at temperatures from -5°C to -25°C, and between 0.1 $L^{-1}$ to $10^5$ $L^{-1}$ at lower temperatures i.e., from -24°C to -38°C."

**line 143-146**: DeMott et al. (2003) even showed that INP from the Sahara were found over Florida.

A/ The reference was added in this section.

**line 158-159:** Do you know the study by Schnell & Vali (1976)? They already report a dependence of INP concentrations with climate zones, albeit only based on leaf litter -read it and then decide yourself if it fits in here.

A / We thank the reviewer for the suggestion; however, we do not consider that the suggested study properly fits with our manuscript. We prefer to focus on other pioneering studies such as Rosinski et al. (1987) and Rosinski et al. (1988) that better align with the scope of the present study.

**Table 1** and related text: CPCs are mentioned in Table 1 but not in the text. If they were used for the here presented analysis, add a description to the text. If they were not used in this study, delete them from the table. Also: Which CPC was used in Merida in 2017? Please specify in the table! Also: Mini-Vol was an abbreviation used in the text, in the table it's MiniVol. Unify!

A/ The CPC was removed from Table 1 as it was not used in the present study. The word "MiniVol" was re-written in the text as suggested.

**line 196** and **line 369-370**: This is one of my main concerns: How were the samples included in this study chosen? Randomly? Or for which other reasons? Or were all collected filters included? If not, Did the choice of a subset influence the conclusions drawn from the study? As I understand, you just attribute different air masses to different seasons? Is that justified? There is a study by Wex et al. (2016), in which within one month on Barbados different air masses were observed (clean marine, marine with new particle formation, and African Dust), all advected across the Atlantic. That was the case in November, and again in April. Similarly, a certain air mass likely cannot be expected to be connected to a certain season at your sampling locations. Therefore: It at least has to be mentioned how you defined/chose the different air masses, and how you checked if all air masses during a sampling time belonged to the type of air mass you expect during this season.

A/ The Yucatan Peninsula is affected by cold fronts between November and March, by biomass burning emissions from Central America and South Mexico between January and May, and by African dust between June and August. Therefore, different samples were collected on each of the abovementioned periods of interest. Although several samples were collected during each sampling period for ice nucleation analysis, Table S1 lists the samples where ice nucleation for the full size range was available (i.e., from 0.32 $\mu$m to 10 $\mu$m). During sampling and data analysis some samples got damaged, and therefore, the full size range was incomplete. Also, in some cases, too many particles impacted in the substrates which inhibits the proper formation of water drops. On those samples the ice nucleation experiments could not be performed, leaving the full size range also incomplete. In summary, the entire study and the drawn conclusions are based on the samples shown in Table S1.

The characteristics of each air mass were not simply assigned by the different seasons. The chemical composition of the aerosol particles collected during the MA, BB, and AD air masses are clearly different and in accordance with the literature (Figure 5). Also, the particle size distribution of each sampling period was monitored

as shown in Figure 6. Finally, Figure S3 shows that the air mass histories for the MA, BB, and AD are very distinct.

**line 215-216**: How much time passed between sampling and analysis? Give that number here. Have you ever tested if the INP concentration is influenced by storage at temperatures above 0_C? Most studies I am aware of freeze their samples prior to the analysis.
A/ Given that the aerosol sampling took place before the UNAM-MOUDI-DFT was fully operational and calibrated, the time that passed between the sampling and the ice nucleation analysis varied between 3 and 22 months. Note that it is very common to store the samples at 4°C (or even room temperature) when using the MOUDI-DFT (e.g., Mason et al. 2015; DeMott et al 2016; Mason et al. 2016; Si et al. 2018). Although in the previously mentioned studies the time that the samples were stored in the fridge did not exceed 7 months, Figure A3 shows that the frozen fraction for the biomass burning samples collected in (a) 2017 and (b) 2018 have a similar behavior. Note that the 2017 samples were stored for 24 months, while the 2018 samples were stored for 12 months.

[Figure]

Figure A3: Frozen fraction curves for (a) 2017 and (b) 2018 biomass burning samples.

- Mason, R.H. et al.: Ice nucleating particles at a coastal marine boundary layer site: correlations with aerosol type and meteorological conditions, Atmos. Chem. Phys., 15(21), 12547–12566, doi:10.5194/acp-15-12547-2015, 2015.
- Mason, R.H. et al.: Size-resolved measurements of ice-nucleating particles at six locations in North America and one in Europe, Atmos. Chem. Phys., 16(3), 1637–1651, doi:10.5194/acp-16-1637-2016, 2016.
- DeMott, P.J., et al.: Sea spray aerosol as a unique source of ice nucleating particles, Proc. Natl. Acad. Sci. U. S. A., 113(21), 5797–5803, doi:10.1073/pnas.1514034112, 2016.
- Si, M., et al.: Ice-nucleating ability of aerosol particles and possible sources at three coastal marine sites, Atmos. Chem. Phys., 18(21), 15669–15685, doi:10.5194/acp-18-15669-2018, 2018.

**line 370ff**: You show in Fig. 5 and the related text, that "background conditions" were different from other days during the three different air mass types (particularly pronounced, to my understanding, for AD). How was this variability treated when you then summarized the results to obtain one value representative for the whole air

mass (as done for Fig. 6 and Fig. 8)? This is, again, linked to how you chose the samples that are presented here.

A/ The background was defined as the lowest $PM_{2.5}$ and $PM_{10}$ values within each sampling period where the chemical composition was available. This is mentioned in the revised manuscript.

Lines 437-439: "The background days were defined as those with the lowest $PM_{2.5}$ and $PM_{10}$ values within each sampling period when the chemical composition was available."

Figure 6 does not relate to chemical composition as it illustrates the particle size distribution obtained from the optical particle counter (LasAir). Regarding Figure 8, it combines the INP concentration and the surface area derived from the data shown in Figure 6. Therefore, the background (mentioned in lines 370) does not have a direct link to neither Figure 6 nor Figure 8.

**line 392:** Of course, if there is also a marine influence (which is to be expected), then part of the sulfur will also some from the same sources as described for MA above. Reformulate this passage accordingly!

A/ Unfortunately, we do not have the means to differentiate between the different sources of sulfur. However, following the reviewer's advice, the text was modified as follows. Lines 458-462: "Sulfur is the most abundant element observed in Merida under background conditions (Figure 5b, left panel). The presence of sulfur can be attributed to anthropogenic sources, such as light- and heavy-duty vehicles with diesel engines that are prevalent in Merida. Given the proximity of Merida to the Gulf of Mexico, part of the sulfur measured in Merida could originate from dimethyl sulfide (DMS)."

**line 429**: Am I right to assume that you calculated the particle concentration based on the measured size distributions? Please mention explicitly how concentrations were obtained!

A/ Although the optical particle counter (LasAir) directly reports the particle concentration for each size bin, the normalized particle concentration (as reported in Figure 6) was obtained as follows:

$$normalized\ particle\ conc.\ (size\ bin\ i) = \frac{particle\ conc.\ (size\ bin\ i)}{size\ bin\ width\ (size\ bin\ i)}$$

The text was modified for clarity as follows. Lines 497-499: "For all three air masses, the highest particle concentration (reported by the LasAir) was observed for particles with EODs between 0.3 μm and 0.5 μm."

**line 453-454**: There are trade wind cumuli across the Atlantic and they can produce warm rain! Just for fun I opened "windy.com" while writing this, checked for clouds between Africa and Yucatan, and (as I expected and as you will always see) there sure were several locations with intense cloud fields. Please correct this statement!

It is clear, though, that particles are transported in "SAL" (Saharan Air Layer), and yes, "trickling out" from that indeed happens. But wet removal likely occurs, too.

A/ The text was modified as follows. Lines 522-524: "While crossing the Atlantic, many particles can be removed from the atmosphere by dry deposition and, to a lesser extent, by wet deposition as cloud cover is low during this season."

**line 461-463**: Was the study you cite here based on a lab-experiment? If yes, then the dilution of the air also will have played a role, and then it is not clear to me how representative this is for atmospheric concentrations, and this quote might not be very meaningful.

A/ The reviewer is correct. As the study was performed in laboratory conditions, it is not relevant to our manuscript, and therefore, it was deleted.

**line 467**: In Tab. S1, there are 7, 5, and 4+4=8 samples for MA, BB and AD, respectively. The number of data-sets shown in Fig. 7a seems lower. Is that so, and if yes, why?

A/ This is an optical effect. As shown below in Figure A2, when using different colors for each sample, the 20 curves are distinguishable.

**line 471-473**: "onset freezing temperatures" are not a good value for judging a data-set, as they will depend on the amount of air sampled and the instrumentation used for the evaluation. At least add this information, or better delete this sentence completely.

A/ We agree with the reviewer. Therefore, the sentence was deleted from the original manuscript.

paragraphs starting at **475** and at **488**: It is much better to give comparative data in a figure instead of giving a range of numbers in a text. Additionally, a concentration of INP always has to be given at a temperature. Giving a range of concentrations spread over a range of temperatures is not helpful, and comparing concentrations obtained at different temperatures or for different temperature ranges is not meaningful, either.

I suggest that you make an additional figure with your data with separate panels for the three air masses in which you always add the literature data you want to compare to. By reading this, one cannot understand the relation, and with this new figure you can avoid text filled with numbers which is difficult to read. And, as you quite correctly say, you are far away from the source, compared to the other studies to which you are comparing to, so that comparing a concentration is not the best thing to do, as a concentration HAS TO decrease further away from a source. Comparing intrinsic INP properties, such as n_s, makes more sense.

A/ We agree with the reviewer that $n_s$ is the best way to compare different ice nucleation data sets, which is the reason for Figure S4. We feel that a new figure where we add the INP concentrations from the present study and compare them with the literature is not necessary.

**line 489**: Actually, INP concentrations for MA and AD are surprisingly similar, both in the range they cover and in their lowest and highest values (besides for these few

high temperature data points - but here it might be reflected that some samples collected more air volume than others), so your statement here does not reflect that correctly. I am not even sure if the discrimination between MA and AD makes sense? Couldn't there also be dust particles for some of the filters collected during MA? The one trajectory shown in the SI doesn't say much about this.
A/ Please see our answer to Major point (3).

The total volume of a given sample should not impact the INP concentration as we are reporting the INP concentration per liter of air. This means that each sample is normalized by its corresponding volume.

The discrimination of MA and AD is clearly detailed and supported in our manuscript. Note that between January and February (MA period) the chance that an air mass from Africa reaches Mexico is extremely low. At that time of the year, the air masses leaving the African continent towards the Atlantic move southward reaching South America (Kischa et al. 2014). Finally, the impact of the physicochemical properties of the aerosol particles collected in MA and AD on ice nucleation is evident in Figure S4, where the $n_s$ values obtained for both types of aerosol particles is clearly different.

- Kishcha, P., da Silva, A. M., Starobinets, B., Long, C. N., Kalashnikova, O., & Alpert, P. (2014). Meridional distribution of aerosol optical thickness over the tropical Atlantic Ocean. Atmospheric Chemistry and Physics Discussions, 14, 23309-23339.

Mineral dust particles are ubiquitous in the continents, and therefore, they cannot be completely ruled out. However, based on the clear differences of the MA and AD chemical composition shown in Figure 5, and the clear absence of Fe, Al, and Si during the MA period, we are confident that the source of aerosol particles collected during the MA and AD periods are completely different.

**line 511**: Why is the trajectory shown for BB in the SI so short? If African dust would have been sampled during BB, that could dominate what you see. Alternatively, it could explain that the data is quite low if BB would not have contributed much to INP and the air mass did not come from Africa. A longer trajectory can help the discussion.
A/ The BB trajectory was run for a shorter time because the BB emissions arriving at Merida are likely coming from fires located nearby, i.e., the southeastern region of the peninsula and Central America. Rios and Raga (2018) present the spatial distribution of fires in the region using 14 years of data, as seen in Figure A4.

[Figure]

Figure A4. Spatial distribution of 14-year average of burned area (km²) in Mexico and Central America for the months of January- June.

Smoke plumes from these regional fires reach the sampling site in only a few hours. They are not advected from Africa and there is no need for longer backtrajectories. Note also that June has very few fires in southern Mexico and Central America, since the rainy season has already started.  As explained in the previous answer, African dust can effectively reach Mexico between June and August (not in April), during the relative minimum in precipitation observed yearly associated with an intensification of the Caribbean Low Level Jet. Again, we invite the reviewer to check the clear differences in aerosol chemical composition during the AD and BB periods (Figure 5).

**line 519**: "(likely southern Mexico and Central America)" again made me wonder how you defined periods which were included in BB.
A/ As shown in Figure A, there is a clear BB season in Mexico and Central America with the highest burned areas found between April and May (Rios and Raga, 2018). Note that neither Cold fronts, nor African dust intrusion takes place in April (when the BB samples were collected).

[Figure]

Figure A5. Annual cycle of burned area (in km2) over the period (2001–2014) in Mexico and Central America, as a function of the various land-cover types (Rios and Raga, 2018).

- Ríos, B. and Raga, G. B.: Spatio-temporal distribution of burned areas by ecoregions in Mexico and central America, Int. J. Remote Sens., 39(4), 949–970, doi:10.1080/01431161.2017.1392641, 2018.

**equation 3 and related text**: Was the total surface area based on the aerodynamic diameter at the sampling RH used, here? From the parameter-names you use in the main text and in the SI, it seems that this is not the case? Please explain all this in more detail, at least in the SI. If the aerodynamic diameter at the sampling RH was used to calculate n_s, then why? Typically, dry particle sizes are used. It is not described anywhere, why all these transformations are given in the SI for the calculation of surface active site density (n_s) (aerodynamic to geometric, dry to RH). This certainly has to do with your optical size measurement, but all of this needs to be made clearer.

A/ We apologize for the confusion in this section. To calculate the $n_s$ values, the particle concentrations were obtained from the optical particle counter LasAir. Therefore, in our analysis the equivalent optical diameter (directly reported by the LasAir) was used without any conversion and without the usage of the aerodynamic diameter. The text was modified to properly reflect the way $n_s$ was calculated.

**line 541ff**, related to Fig. S4: Umo and Grawe only looked at ash particles, so their n_s relates to these, only. You include the surface area of all other particles, too, therefore it is logical that you end up with lower values. This should be said somewhere explicitly in order not to confuse the readers. Something similar is true for most of the studies you used here for comparison: They examined a separate substance. Even Price only used the surface area of the (massive) amounts of dust

particles, if I am not mistaken. It therefore makes perfect sense that mainly Gong fits as all others did not determine the n_s of an air mass with a mix of particles in it, but the n_s of one particle type. Also: are these separate components (starting with montmorillonite) your own measurements? Or else where do they come from? Adjust the text in the main manuscript accordingly!

A/ We are aware that our samples have a mixture of particles (as do most ambient samples). However, our intention when carrying out this analysis is to have a reference point with previous studies to infer if the samples collected at this tropical site are more (or less) efficient at nucleating ice. Note that several assumptions are made when calculating $n_s$, with the particle density ($\rho_p$) being one of the most important. Typically, a representative $\rho_p$ value is used for ambient samples as it is impossible to determine the $\rho_p$ for each aerosol type present in mixed air masses.

**line 582-583**: As said above, INP concentrations of MA and AD were very similar as presented in Fig. 7 (as said, do not overinterpret the onset-temperatures), while indeed n_s was different. But are the INP in MA and SD clearly different? It's fine if this can be shown, but after going through the reviews, please adjust the text here accordingly.

A/ We agree with the reviewer that the INP concentrations shown in Figure 7 look rather similar for all three air masses. However, as shown in Figure A2 (above) there are significant differences within the three aerosol types. We do not think that we are overinterpreting the onset freezing temperatures. We are not comparing the ice nucleating abilities of the three air masses based on the onset freezing values, we did it based on the $n_s$ values. In lines 582-582 (original manuscript) we are simply reporting what was observed in our study.

**Technical comments:**

**line 39-40**: Add the temperatures at which the INP concentrations given here are valid - without that, it cannot be judged if these are high or low.

A/ The corresponding freezing temperatures were added as follows: "AD (from 0.071 $L^{-1}$ to 36.07 $L^{-1}$ at temperatures between -18°C and -27°C), followed by MA (from 0.068 $L^{-1}$ to 18.90 $L^{-1}$ at temperatures between -15°C and -28°C), and BB (from 0.063 $L^{-1}$ to 10.21 $L^{-1}$, at temperatures between -20°C and -27°C)."

**line 61**: Change "its" to "their".

A/ Thank you. Corrected.

**line 78**: "Augustin-Bauditz et al. (2008)": I checked, as the date struck me as quite early, but this is a publication from 2014. This study cited here also focusses on Kfeldspar as being the main mineral that drives ice nucleation, so it could additionally fit to what is said in line 84. (Discovery of this slip caused that I randomly checked other citations as well - Welti et al. (2020) should be cited as given below, and in the SI, at least Grawe et al. (2016) is missing in the References. - There may be other issues, please check.)

A/ We apologize for these mistakes. The reference was corrected to "Augustin-Bauditz et al. (2014)"

*The reference of Grawe et al. (2016) was added to the SI.

"Grawe, S., Augustin-Bauditz, S., Hartmann, S., Hellner, L., Pettersson, J. B. C., Prager, A., Stratmann, F. and Wex, H.: The immersion freezing behavior of ash particles from wood and brown coal burning, Atmos. Chem. Phys., 16(21), 13911–13928, doi:10.5194/acp-16-13911-2016, 2016."

*As the Welti et al., (2020) was accepted the reference was modified as follows:

"Welti, A., Bigg, E. K., DeMott, P. J., Gong, X., Hartmann, M., Harvey, M., Henning, S., Herenz, P., Hill, T. C. J., Hornblow, B., Leck, C., Löffler, M., McCluskey, C. S., Rauker, A. M., Schmale, J., Tatzelt, C., van Pinxteren, M., and Stratmann, F.: Ship-based measurements of ice nuclei concentrations over the Arctic, Atlantic, Pacific and Southern Ocean, Atmos. Chem. Phys., https://doi.org/10.5194/acp-2020-466, in press, 2020."

**line 91-92**: The samples for this study were not taken on Cape Verde, but flying over the ocean around Cape Verde, with some flights closer to the African continent. Maybe say "... and -23_C for airborne samples collected under dusty conditions around Cape Verde, Africa , ...".
A/ "around" was added.

**line 94-95**: If you follow the above advise, then here "in the same region" has to become "on one of the Cape Verde islands."
A/ We thank you for the suggestion. This was fixed.

**line 120**: Delete the second "the" (at "in the Arctic the").
A/ Thank you. Removed.

**line 163**: Add "of the role" between "understanding" and "that".
A/ The sentence was modified as follows. Lines 200-201: "there is an urgent need to improve the current understanding of the role of aerosol particles, emitted in tropical latitudes, in cloud formation".

**line 190**: Concerning "20_C and 34_C": What are these two different temperatures (different seasons? day and night?)? Why are there two values here while above it was only one value for Merida and the Yucatan peninsula?
A/ The sentence was modified as follows. Lines 229-230: "The mean temperature and RH are 27°C and 80.8%, respectively."

**line 191**: For the RH, the "+" part here makes no sense, as that ends up well above 100%.
A/ The sentence was modified as follows. Lines 229-230: "The mean temperature and RH are 27°C and 80.8%, respectively."

**line 196**: Exchange "for presented" with either "for presentation" or "to be presented".

A/ Thank you. Corrected.

**line 228**: Better use "consists of" instead of "is integrated by".
A/ Thank you. Corrected.

**line 230** and **line 232** and **line 236**: cmˆ3 (not only cm)!
A/ We are reporting the dimensions of the cell and blocks and not the volume. This was fixed.

**line 235**: Where is this thermocouple? Maybe explicitly hint to Fig. S1 here, too.
A/ Thank you. This is now indicated in Figure S1.

**line 251**: How is the air "generated"? Maybe better: "added to the nitrogen flow when it is conducted ...".
A/The text was modified as follows. Lines 290-291: "dry conditions are experienced when nitrogen is injected and flows towards the sample".

**line 255**: Do you know the temperature variation across the glass plate? Please add this information here.
A/ Although we assume that the temperature across the glass plate is constant, we did not evaluate this as the experiments are performed in a very small area (1.2 mm$^2$). The RTD is placed right below the area we analyze in the DFT.

**line 261**: Be careful with the time form you use - this whole paragraph is in present tense, besides for this "were", here. -> are.
A/ Thank you. Fixed.

**line 261**: Instead of "with" better "on" or "in".
A/ Thank you. Fixed.

**line 296**: Either " diameters ranging from ... to ..." or "diameters between . . . and . . .".
A/ Thank you. The word "ranging" was deleted.

**line 297**: Give the size range for which these two LasAir-instruments measured, in the text or table.
A/The text was modified as follows. Lines 351-353: "The particle size distributions for equivalent optical diameters (EOD) between 0.3 $\mu$m to 25 $\mu$m were obtained with optical particle counters, a PMS LasAir II 310A (0.3 $\mu$m to 25 $\mu$m) and PMS LasAir III 310C (0.3 $\mu$m to 10 $\mu$m)."

**line 325**: Why do you give two different lengths for the trajectories? Different for different samples / locations/ . . . ? Explain this in the text.
A/ There was a typo in this line as the back trajectories were run for 24 h and 13 days. The text was modified as follows.

**Line 386-389:** "The 24 h back trajectories were run for the BB events as they were likely locally emitted. On the other hand, the back trajectories for the cold fronts (MA) and African dust intrusions (AD) were run for 13 days as the particles carried by those events take several days to reach Mexico."

**line 402**: Replace "areas" with "parts".
A/ Replaced as suggested.

**line 402**: Add "is" prior to "positively".
A/ We think the grammar is correct here.

**line 404**: Add "a comparatively high mass fraction of" between "contained" and "particles".
A/ Thank you. Your suggestion was added.

**line 427**: Fig. 6 shows the concentration, not the distribution -> correct.
A/ The data shown in Figure 6 is actually an aerosol particle size distribution. It shows how the concentration of aerosol particles changes as a function of their size.

**line 487**: At least mention briefly, what the explanation is, that is given for your observation in the here cited publication.
A/ The text was modified as follows. Lines 556-559: "Therefore, the high wind speeds associated with the cold fronts, and hence wave breaking, could have played an important role in the high INP concentrations measured in Sisal, as discussed in Ladino et al. (2019)."

**line 558:** Concerning "mixed-phase cloud formation": this should better be "immersion freezing" - you do not mimic the formation of a cloud, that would be up to cloud chambers.
A/ The sentence was modified as follows. Lines 631-633: ". This new device can be used to study ice formation via immersion freezing and can discriminate the INP concentrations as a function of the aerosol aerodynamic size from 0.32 $\mu$m to 10 $\mu$m"

**line 561-564**: As that PICNIC-study is not a topic described in your manuscript (besides that you mentioned that this other study exists), this should not be part of these conclusions here. Sentence should be deleted.
A/ We respectfully disagree with the reviewer. Given that the present study also covers the performance of the UNAM-MOUDI-DFT, we are convinced that this sentence can serve as a guide to the readers interested in our device. Similar statements can be often found in the conclusions of scientific publications.

**line 576**: Concerning "and also at tropical latitudes." - How about Gong et al. (2020), which you've cited a number of times?

A/ Gong et al. (2020) was indeed performed at Tropical Latitudes; however, the authors focused on a single aerosol type (MA), instead of three different and distinct air masses as in the present study.

**Fig. 4**: In the caption: "Khon" -> "Kohn".
A/ Thank you. This was fixed.

**Fig. 7b**: Black writing in a dark blue field is quasi invisible - change color of the text to white.
A/ Thank you. This was fixed.

**SI, Fig. S3**: Typically the sequence you used so far is "MA", "BB", "AD" (following the sequence with which you sampled), so it is a bit confusing that the latter two are swapped, here. Please exchange.
A/ Thank you. This was fixed.

**SI, line 89**: "base" -> "based".
A/ Thank you. This was fixed.

---

## Referee Report (RR1)

Review of "Measurement report: Ice nucleating abilities of biomass burning, African dust, and sea spray aerosol particles over the Yucatan Peninsula" by Córdoba et al.

**General comment**

The authors provided satisfactory answers to the comments which were previously raised, and therefore I recommend its publication after correcting the following technical issues:

**Technical comment**

L121: "Mccluskey" should be "McCluskey".Please change along the text.

L138: Remove one space after the word: "Vancouver"

L246: A space is missing between cm and x.

L537: "eastern" should be "Eastern"

L640: LL is LAL?

---

## Author Response (AR2)

We would like to thank the Editor and the Reviewers for their additional constructive suggestions, which helped us to improve the manuscript. Specific answers and manuscript modifications related to the Editor and Reviewers comments are given below in red text.

**Editor**

**(1)** Referee 2 comment 1: I agree with referee 2 that it appears somewhat unusual to define the three types of aerosols (marine aerosol, MA, biomass burning, BB, and African dust, AD) by the time period of sampling, rather than by their chemical composition or by their origin (via back-trajectory analysis). I do understand that there is a correlation between those classifications, but from the data provided in the current manuscript, it is not clear to me how strong this correlation is. Such information is required! Moreover, you have defined background aerosol number and composition. Apparently, these are distinctly different from the general classification aerosol properties (see Figs5b and 5c in particular). So my question: Is the background day just one unusual day within the time-period otherwise influenced by the general aerosol according to its classification, or do "background days" occur more often? Note that because of the high non-linearity of ice-nucleation activity between different types of aerosol, the time-period averaging will strongly depend upon the fraction of days with aerosols of the general classification during that period. It will also depend upon the fraction of INP measurement days dominated by that particular class. For that purpose, I suggest to make that classification based on chemical analysis for the days of INP measurements or, if this is not possible because of non-sufficient overlap between days with chemical analysis and days of INP measurement, based on back trajectories for the INP measurement days. Then do provide the numbers for the fractions of days in each period at least. Moreover, I appreciate the addition of supplementary table S1. However, even when considering both Table 1 and Table S1, I found myself struggling on which days INP data, size distribution data, and chemical analysis data were obtained. A more detailed overview table with individual days of INP measurements, days of chemical analysis measurements and the definition of aerosol classifications would help. All readers interested in further detail can then have a look at the original data of the individual analyses and INP measurement in the deposited original data, see point (7) below.

A/ We thank the Editor and the Reviewer for pointing out the lack of clarity in the discrimination between the three air masses and the meaning of the backgrounds.

Prior knowledge of the seasonality of BB and AD events was crucial to select the periods to carry out the field campaigns. As shown below in Figure A1, April is the time of the year where the maximum fire density is found in the Yucatan Peninsula. Note, that these results were obtained from a 14-year study using satellite information (Rios and Raga, 2018).

[Figure]

Figure A1. (Top) Spatial distribution of the 14 year average burned area (km²) in Mexico and Central America for the period (2001–2014) between January and June. (Bottom) Spatial distribution of the 14 year average burned area (km²) in Mexico and Central America for the period (2001–2014) between July and December (Ríos and Raga, 2018).

Likewise, Figure A2 shows that July has the highest probability of dust arrival in the Yucatan. Moreover, July is characterized in many regions around the Caribbean as part of the mid-summer drought, when there is a reduction in precipitation and stronger trade winds and the presence of the Caribbean Low Level Jet, which has been studied since the 90s (e.g. Magaña et al, 1999). Note that Figure A2 was obtained from 20 years of reanalysis using MERRA-2.

[Figure]

Figure A2. Mean monthly PM₁₀ concentration associated with dust derived from MERRA-2 over 2000-2019, for June through September. The Yucatan peninsula is located in southeastern Mexico, surrounded by the western Caribbean to the East and the Gulf of Mexico to the North and West (Raga et al. 2020).

To clarify how the MA, BB, and AD periods were chosen, the following text was added to the revised manuscript. Lines 216-228: "*The presence of BB particles and the intrusions of AD onto the Yucatan Peninsula has been previously documented (e.g., Yokelson et al. 2009; Kishcha et al. 2014; Rios and Raga, 2018; Raga et al. 2020; Trujano-Jiménez et al. 2021; Ramirez-Romero et al. 2021). Rios and Raga (2018) reported that within the BB season, the maximum fire density is observed in April. Likewise, Kishcha et al. (2014), Raga et al. (2020), and Ramirez-Romero et al. (2021) indicate that July is the period with the highest likelihood for the AD influx to the Western Caribbean within the (MSD). Therefore, April and July were chosen as the sampling periods to capture the presence of BB and AD particles in Mérida, respectively.*

*Given that the Yucatan Peninsula is encircled by the GoM, MA is ubiquitous throughout the Peninsula. Therefore, the MA composition was assessed in the remote coastal village of Sisal between January and February, a time of the year where the presence of particles such as BB and AD are least likely.*"

Having in mind that the aerosol particles in the Yucatan Peninsula in January, April, and July are heavily influenced by MA, BB, and AD, respectively (based on the literature), we analyzed the aerosol's chemical composition and the history of the air masses, to corroborate their origin.

Regarding Figure 5, we would like to mention that the original figure showed the elemental composition for several days, from each period, as examples for typical days. The background compositions were added to left of those panels in order to highlight the differences for the readers. However, as the figure has created confusion, we have modified it (as shown below) and subsection 3.2 was rewritten (Lines 431-498). The redrawn Figure 5 shows the background composition by site rather than by sampling period to avoid confusion and the influence of the BB and AD in the aerosol composition was assessed by adding the enrichment factor for each of the 16 analyzed elements.

[Figure]

**Figure 5.** Percentage elemental mass concentration for background samples in Merida (panel a) and Sisal (panel b). Enrichment factors are calculated for samples obtained under the influence of BB (red bars) and AD (yellow bars) plumes from background conditions in Merida (panel c).

As mentioned above, the data provided in the original Figure 5 corresponded to some example days and not to the full data available. We may have given the impression that we only have the chemical composition for limited days; however, we would like to clarify that the chemical composition was obtained for >90% of those days where the INP concentrations are being reported in the present study. We hope that the new Figure 5 (and the corresponding text) has alleviated the reviewer's and editor's concerns.

The aim of Figures 5 and S3 was to provide robust evidence that the aerosol particles collected in January, April, and July were different in origin. In parallel to the aerosol particles collected to analyze their chemical composition, aerosol particles were collected with the MOUDI to evaluate their ice nucleating abilities. Therefore, we have INP measurements for each sampling period together with chemical composition and size distribution as shown below in the revised Table S1.

**Table S1.** Summary of a subset of samples (those with the full size range available i.e., from 0.32 μm to 10 μm) taken from Merida and Sisal during 2017 and 2018 to analyze the results presented in this study. MA, BB, and AD refer to marine aerosol, biomass burning, and African dust, respectively. The last two columns indicate if the chemical composition and the size distributions were available parallel to the INP samples. *two samples were collected at different times during the same day.

| Aerosol Type | Place | Date | Chemical composition available | Size distribution available |
|---|---|---|---|---|
| MA | Sisal | 24-01-2017 * | Yes | Yes |
| | | 24-01-2017 * | Yes | Yes |
| | | 25-01-2017 | Yes | Yes |
| | | 26-01-2017 | Yes | Yes |
| | | 27-01-2017 | Yes | Yes |
| | | 28-01-2017 | Yes | Yes |
| | | 29-01-2017 | Yes | Yes |
| BB | Merida | 27-05-2017 | No | No |
| | | 03-04-2018 | Yes | Yes |
| | | 05-04-2018 | Yes | Yes |
| | | 07-04-2018 | Yes | Yes |
| | | 08-04-2018 | Yes | Yes |
| AD | Sisal | 11-07-2018 | Yes | Yes |
| | | 12-07-2018 | Yes | Yes |
| | | 13-07-2018 | Yes | Yes |
| | | 15-07-2018 | Yes | Yes |
| | Merida | 11-07-2018 | Yes | No |
| | | 13-07-2018 | Yes | No |
| | | 14-07-2018 | Yes | Yes |
| | | 16-07-2018 | No | No |

**(2)** Referee 2, comment 2: "Only one in 10^5 to 10^6 of the aerosol particles can act as INP at temperatures higher than -38°C (Lohmann et al., 2016)." I agree with you that this is a cited statement from the given reference. Considering the comment by the referee, I suggest that you may consider making this sentence less strict and approximate.

*A/ Lines 69-70 were changed as follows "One in $10^6$ (or fewer) of atmospheric aerosol particles can act as INP at temperatures higher than -38°C".*

**(3)** Referee 2, comment 3: I agree with the suggestion of the referee for changing the text to: "This is the first such comprehensive study ever conducted in Mexico and among the first ones at tropical latitudes." Or something similar.

*A/ Thanks for your suggestion. The text was modified as follows. Lines 659-660: "This is the first such comprehensive study ever conducted in Mexico and among the first ones at tropical latitudes."*

**(4)** Referee 2, comment 4: I agree with the referee that a statement on storage time of the samples should be added to the experimental part, and in that one you may also make a statement, as you did in your previous answer to the referee, that you did not observe differences between 12 and 24 months of storage. Moreover, the current study was performed before the Beall et al. (2020) study on protocols for the storage of INP samples became publically available, and it was submitted before the latter was accepted for final publication. Therefore, I think it suffices to only briefly mention that study in that respect.

Thank you for your suggestion. The text was modified as follows. Lines 245-250: *"After each sampling, the glass coverslips were stored in Petri dishes between three and twenty-two months at 4°C prior to their analysis with the DFT in Mexico City (Fig. 2a). As recently highlighted by Beall et al. (2020), the temperature and the length of the storage can impact the ice nucleation abilities of MA samples. Although, this was not evaluated in the present study, future studies will evaluate how the storage procedure impacts results."*

**(5)** Referee 2, comment 5: While the referee is correct in their statement, I think it is okay if you leave the data by Umo et al. (and other data sets) in the comparison figures. But I would ask you to add a statement on that account to inform readers, i.e. "Note that some of the n_s data in the comparison refer to pure components or aerosol types, while our analysis includes the entire atmospheric aerosol variability, i.e. also the surface on particles not acting as ice nucleating particles" or something along this line.

*A/ Thank you for the suggestions. The following text was added to the revised manuscript. Lines 613-615 "Note that some of the $n_s$ data in the comparison refer to pure components or aerosol types, while our analysis includes the entire atmospheric aerosol variability, i.e., also the surface of particles not acting as ice nucleating particles."*

**(6)** Referee 2, comment 6: I agree with the referee, so please add the information on how the size distribution for calculating the n_s values were obtained/calculated so that it can be clearly understood and repeated.

*A/ This section was modified to include the requested information as follows:*

**"Calculation of surface active site density ($n_s$)**

*The methodology employed in this study is based on Si et al. (2018). First, the particle's density was calculated at a given RH ($\rho_{p,RH}$) using Equation S_E1. Later on, x factor was calculated following the Equation S_E2. Note that the x factor was computed for each air mass type (MA, AD, and BB). Finally, to obtain the $n_s$ values using Equation S_E3, the particle concentration as a function of their size measured with LasAir and the INP concentrations were necessary. The $n_s$ was obtained for each individual sample listed in Table S1, and therefore, the INP concentration and the particle concentration of each individual sample (and for each period) were used.*

1. **Calculation of the particle density at a given RH ($\rho_{p,RH}$)**

$$\rho_{p,RH} = \rho_w + (\rho_{p,dry} - \rho_w)\frac{1}{gf^3} \qquad (S\_E1)$$

*where $\rho_w$ is the density of water and $P_{p,dry}$ is the density of the dry particles. 1.87 g cm$^{-3}$ was used for marine aerosol (Si et al., 2018), 2.5 g cm$^{-3}$ for mineral dust particles (Wheeler et al., 2015) and 1.25 g cm$^{-3}$ for biomass burning particles, as it is the average between 1.1 g cm$^{-3}$ and 1.4 g cm$^{-3}$ reported by Li et al. (2016). $gf^3$ is the hygroscopic growth factor that was obtained from Ming and Russell (2001), using the mean relative humidity for Sisal in January (65%) and July (95%), and for Merida in April (65%) and July (90%). The particles were assumed to be composed of 30% of organic species.*

2. **Calculation of factor (x).**

$$x = gf\sqrt{\frac{\rho_{p,RH}}{\chi\rho_0}} \qquad (S\_E2)$$

*where $\chi$ is the dynamic shape factor for a non-spherical particles and $\rho_0$ the unit density of 1 g cm$^{-3}$.*

3. **Calculation of $n_s$ based on the geometric diameters at a given RH.**

$$n_{s\_ae\_RH} = \frac{[INP]}{S_{tot,ae,RH}} = \frac{[INP]}{\pi x^2 D_{geo,dry}^2 N_{tot}} \qquad (S\_E3)$$

*where [INPs] is the concentration of INP (L$^{-1}$) at each temperature (i.e., -15°C, -20°C, -25°C, and -30°C) for each sample. $S_{tot,ae,RH}$ is the total surface area based on the aerodynamic diameter at the sampling RH, and $N_{tot}$ the total number of aerosol particles. $D_{geo, dry}$ corresponds to the average diameter of each LasAir size bin as shown in Table S2. In this case, only the data from channels 1 to 4 from the LasAir were used since these size range overlaps with the MOUDI diameters from stage 2 to 7.*

*Table S2. LasAir channels and diameters used in the present study with their corresponding surface area.*

| LasAir channel | Diameter range ($\mu m$) | MOUDI stages ($\mu m$) | $D_{geo, dry}$ ($\mu m$) | $D^2_{geo, dry}$ ($cm^2$) |
|---|---|---|---|---|
| D1 | 0.3 – 0.5 | 7 (0.32-0.56) | 0.4 | 1.60 x10$^{-9}$ |
| D2 | 0.5 – 1.0 | 6 (0.56-1.00) | 0.75 | 5.62 x10$^{-9}$ |
| D3 | 1.0 – 5.0 | 3+4+5 (1.00-5.6) | 3.0 | 8.99 x10$^{-8}$ |
| D4 | 5.0 – 10 | 2 (5.6-10.0) | 7.5 | 5.62 x10$^{-7}$ |

*For each sample listed in Table S1, the INP concentration was derived. Also, given that the LasAir continuously measures the particle size distribution (PSD), the total particle concentration ($N_{tot}$) for each LasAir size bin listed in Table S2 was derived for exactly the same time interval of each INP sample.*

*As shown in Table S2, the INP concentration from the MOUDI stages 3, 4, and 5 were combined in order that their sizes match those from the LasAir size bins when calculating the $n_s$ values for each sample."*

**(7)** As this manuscript is being considered as a measurement report, the original data of the study has to be made available, e.g. either as data sets in the supplement, or deposited and archived in public databases (with doi or hyperlink given): "The data presented in measurement reports must be openly accessible in accordance with the EGU data policy." See: https://www.atmospheric-chemistry-and-physics.net/about/manuscript_types.html The data policy statement includes: "Therefore, Copernicus Publications requests depositing data that correspond to journal articles in reliable (public) data repositories, assigning digital object identifiers, and properly citing data sets as individual contributions. […] In addition, data sets, model code, video supplements, video abstracts, International Geo Sample Numbers, and other digital assets should be linked to the article through DOIs in the assets tab." and can be found here: https://www.atmospheric-chemistry-and-physics.net/policies/data_policy.html A very recent example that was published yesterday is shown here: https://acp.copernicus.org/articles/20/15969/2020/

A/ As discussed in the Ice Nucleation Colloquium presented in January 12, 2021, there is a paucity of ice nucleation data from Tropical latitudes and we have been working hard to partially fill this lack of data and to build a unique data set in this direction. Our next step is to develop a "Tropical" ice nucleation parametrization to be implemented in a regional climate model where it can be compared against the currently available ice nucleation parametrizations which mostly contain mid- and high-latitude measurements. Therefore, we are not comfortable making our unique data set available to the wider community before we have developed the aforementioned parametrization. While we are open to sharing our data with interested researchers, we kindly request the Editor to allow us to share our data upon request to the corresponding author in order that we are informed on how our measurements will be used.

**(8)** Please consider the technical/minor points raised by referee 2.

A/ The technical points were corrected.

**(9)** Please correct all the technical points raised by referee 1.
A/ The technical points were corrected.

**Reviewer #1**

**L121**: "Mccluskey" should be "McCluskey".Please change along the text.
A/ Thank you. This was corrected along the text.

**L138**: Remove one space after the word: "Vancouver"
A/ Corrected.

**L246**: A space is missing between cm and x.
A/ Corrected.

**L537**: "eastern" should be "Eastern"
A/ Corrected.

**L640**: LL is LAL?
A/ Corrected.

**Reviewer #2**

The manuscript improved some, and as much as I would like to give the article a "go", I still cannot do this.
The following issues cannot be left uncommented as they are wrong or inconsistent. In the end, the Editor will have to decide, to which extent the authors will have to review their manuscript once more. In any case, below are my remaining concerns, which were mentioned before and not regarded in the revised version.
Therefore I still have to say "major revisions". Most issues can probably be revised with small additions, but with my choice of "major revisions" I want to stress that these changes are really needed and should not be swept aside.
In the following, line numbers refer to those of the following file: "acp-2020-783-author_response-version1.pdf
**1** One of the shortcomings I still see is the way how the data were split into three different periods, referred to as "marine aerosol" (MA), "biomass burning" (BB) and "African dust" (AD). This needs to be explained and motivated somewhat better in the text. I mentioned that before, but as it was not done, I elaborate here, why that should be done, and how the current text is not sufficient:
For each period, it is assumed that INP for each day/sample are representative for the whole period, as no further discrimination is done. In the answers to my previous review, you repeatedly refer to Figs. 5 and S3. These figures show elemental composition for 3 different days for each period (one always referred to as "background") and 3 single back-trajectories, one for each period, respectively.
A/ We thank the reviewer for pointing out the lack of clarity in the discrimination between the three air masses and the meaning of the backgrounds.

Prior knowledge of the seasonality of BB and AD events was crucial to select the periods to carry out the field campaigns. As shown above in Figure A1, April is the time of the year where the maximum fire density is found in the Yucatan Peninsula. Note, that these results were obtained from a 14-year study using satellite information (Rios and Raga, 2018).

Likewise, Figure A2 (above) shows that July has the highest probability of dust arrival in the Yucatan. Moreover, July is characterized in many regions around the Caribbean as part of the mid-summer drought, when there is a reduction in precipitation and stronger trade winds and the presence of the Caribbean Low Level Jet, which has been studied since the 90s (e.g. Magaña et al, 1999). Note that Figure A2 was obtained from 20 years of reanalysis using MERRA-2.

To clarify how the MA, BB, and AD periods were chosen, the following text was added to the revised manuscript. Lines 216-228: "*The presence of BB particles and the intrusions of AD onto the Yucatan Peninsula has been previously documented (e.g., Yokelson et al. 2009; Kishcha et al. 2014; Rios and Raga, 2018; Raga et al. 2020; Trujano-Jiménez et al. 2021; Ramirez-Romero et al. 2021). Rios and Raga (2018) reported that within the BB season, the maximum fire density is observed in April. Likewise, Kishcha et al. (2014), Raga et al. (2020), and Ramirez-Romero et al. (2021) indicate that July is the period with the highest likelihood for the AD influx to the Western Caribbean within the (MSD). Therefore, April and July were chosen as the sampling periods to capture the presence of BB and AD particles in Mérida, respectively.*

*Given that the Yucatan Peninsula is encircled by the GoM, MA is ubiquitous throughout the Peninsula. Therefore, the MA composition was assessed in the remote coastal village of Sisal between January and February, a time of the year where the presence of particles such as BB and AD are least likely.*"

Having in mind that the aerosol particles in the Yucatan Peninsula in January, April, and July are heavily influenced by MA, BB, and AD, respectively (based on the literature), we analyzed the aerosol's chemical composition and the history of the air masses, to corroborate their origin.

Regarding Figure 5, we would like to mention that the original figure showed the elemental composition for several days, from each period, as examples for typical days. The background compositions were added to left of those panels in order to highlight the differences for the readers. However, as the figure has created confusion, we have modified it (as shown below) and subsection 3.2 was rewritten (Lines 431-498).

Also, Figure S3 was modified to include additional days. The initial conclusions did not change, as the trajectories for the three air masses come from different locations. We would like to make clear that the sampling period and the assigned labels MA, BB, and AD are not based on the back-trajectories alone. As explained above, this is based on literature data and previous observations. Therefore, the HYSPLIT back-trajectories were not the sole indicator of air mass source, but rather were complementing a variety of aspects considered jointly.

[Figure]

**Figure S3**. 13-day HYSPLIT back trajectories for the three different air masses for (a) MA-2017, (b) BB-2017, and (c) AD-2018.

However, days for which you show the chemical composition are often not those days for which INP samples were included. For example, of the 7 INP samples you use for MA, the chemical composition is given for two days which were also used for INP analysis, but not for the third. And of the two that are shown one is the background day. For BB and AD, of the three days for which chemical compositions are shown, for each period only one appears among the INP samples.

A/ As mentioned in our previous answer, the data provided in the original Figure 5 corresponded to some example days and not to the full data available. We may have given the impression that we only have the chemical composition for very few days; however, we would like to clarify that the chemical composition was obtained for >90% of those days where the INP concentrations are being reported in the present study (revised Table S1). We hope that the new Figure 5 (and the corresponding text) and the revised Table S1 has alleviated the reviewer concerns and clarified how the decision was made to select specific time periods for sampling.

Also, for these single trajectories that are only shown for each phase (MA, BB and AD), the one single trajectory for BB and that single one for AD are given for days for which no sample is included in the INP analysis. And the trajectory you show for MA is that for the background conditions.

A/ Similar to the original Figure 5, the original Figure S3 showed examples of typical back-trajectories for each sampling period. We have modified Figure S3 to add other days including those where the INPs analysis is being reported.

And what does the background condition mean, anyway? This is nowhere clearly defined (with that I aim at what "background" means with respect to the aerosol during these phases, not that these are the days with the lowest mass concentrations.)
A/ New Figure 5 shows the background composition by site and not by sampling period to avoid confusion. Section 3.2 was rewritten and it includes an explanation on how the backgrounds are defined (Lines 432-499).

It is, therefore, still unclear to me how representative the INP samples you chose are for the conditions during each of the three different phases. This cannot be judged based on these few data you show in Figs. 5 and S3.
A/ The aim of Figures 5 and S3 was to provide robust evidence that the aerosol particles collected in January, April, and July were different in origin. In parallel to the aerosol particles collected to analyze their chemical composition, aerosol particles were collected with the MOUDI to evaluate their ice nucleating abilities. Therefore, we have INP measurements for each sampling period together with chemical composition and size distribution as shown below in the revised Table S1. Although our data set is larger than the listed samples in Table S1, in order that the provided analysis was comparable for the different field campaigns, we did not take into account those samples where the information from "all" MOUDI stages was not available (i.e., from stage 2 (10 μm) to stage 7 (0.32 μm)).

So overall, there is this inconsistency that you treat all days during one period similar when it comes to interpreting INP, but on the other hand you give "background conditions" that are clearly different from other days in the same season. And nowhere is there a complete overview over the conditions for all days for which INP samples were used.
A/ We hope that with the new Figure 5 and the rewritten subsection 3.2 these points were clarified.

Such an overview could quite simply be done if backward trajectories would be given for all days for which INP samples are included, and that is what I recommend you to do.
A/ As mentioned above, Figure S3 now includes additional back-trajectories for each sampling period. However, we would like to stress that the HYSPLIT back-trajectories were not the sole indicator of air mass source, but rather were complementing a variety of aspects considered jointly.

It is also important to mention that our setup (i.e., the UNAM-MOUDI-DFT) is not a high temporal resolution INP sensor as we need to run the MOUDI for >4 h to collect one sample. Therefore, during the >4 h sampling period the origin of the air masses arriving to the sampling site might change. Based on the literature data (see above Figures A1 and A2), the chemical composition of the MA, BB, and AD shown in new Figure 5, and their size distributions (Fig. 6) we are confident that we were sampling aerosol particles from completely different origins in January, April, and July. Thus, we evaluated if those differences in the origin of the aerosol particles, i.e., MA, BB, and AD impacted their ice nucleation abilities as shown in Figures 7, 8 and S4.

Along the same line, it makes no sense to say that this one 24 hour backward trajectory suffices for the BB phase (which is what you answer to my review, but also still say in line 1386 "The 24 h back trajectories were run for the BB events as they were likely locally emitted."); because you later claim, concerning the BB phase, in lines 1589-1591 "The lower concentrations reported in the present study can be attributed to the long distance between the burning areas (likely southern Mexico and Central America) and the sampling site." If that latter is correct (which it may very well be, you cannot exclude this), then the 24 h back-trajectories you showed are too short. If you do the backward-trajectories for all INP samples, use the longer time period for all!

Then, depending on how these trajectories are, the text may have to be adjusted. And if not much adjustment is needed, all the better. But then at least your data put into a much stronger context.

A/ Please see our previous answer.

**2** Line 1073: What really irritated me is, that you write that you understood my concern about the statement "Only one in 10^5 to 10^6 of the aerosol particles can act as INP at temperatures higher than -38°C (Lohmann et al., 2016).", but that you did not want to change it. When something is wrong in literature, it should not be promoted further (even if others have done that before). Agreeing that this statement is wrong but insisting to use it anyway is rather strange to me, particularly as it does not influence anything in your work. Why not saying it correctly? DeMott et al. (2016) more correctly say "... INPs ..., a select subgroup that may represent 1 in 10^6 or fewer of all aerosol particles". Not perfect, but better.

A/ Lines 69-70 were changed as follows "*One in $10^6$ (or fewer) of atmospheric aerosol particles can act as INP at temperatures higher than -38°C*".

**3** Along a similar line as my last comment goes the following, now in lines 1649-1650, "This is the first such comprehensive study ever conducted in Mexico and also at tropical latitudes." I commented that this is not true for tropical latitudes as Gong et al. (2020), a study you cited several times and thus should know about, also measured at tropical latitudes. Your answer was "Gong et al. (2020) was indeed performed at Tropical Latitudes; however, the authors focused on a single aerosol type (MA), instead of three different and distinct air masses as in the present study."

So now we would need to define the word "comprehensive", but in my understanding you are saying "Yes, our statement is not true, but we will make it anyway as this other study did not study the exact same thing as we did."

Additionally, as you say, by now also Welti et al. (2020) is published and includes data for tropical latitudes. I would suggest you change the wording of the respective sentence at least to: "This is the first such comprehensive study ever conducted in Mexico and among the first ones at tropical latitudes."

A/ The text was modified as follows. Lines 659-660: "*This is the first such comprehensive study ever conducted in Mexico and among the first ones at tropical latitudes.*"

**4** Line 1255: Different from what you say, it has been discussed in the community already for quite some time that samples need to be stored frozen. It has even been shown that long storage times under freezing conditions may change a bacterial sample (Polen et al., 2016). It now was reported that storage at -20°C is mandatory (Beall et al., 2020). In your answer to

my first review, you compare two groups of samples which were both stored at 4°C for many months (12 and 24), and their similarity might just mean that both degraded to the same degree, and that after many months of storage at this comparably high temperature of 4°C no further changes occur. At least cite and shortly comment on the new study, as this very well could influence the outcome of your study. And consider changing your procedures in the future.

A/ Thank you for your suggestion. The text was modified as follows. Lines 245-250: "*After each sampling, the glass coverslips were stored in Petri dishes between three and twenty-two months at 4°C prior to their analysis with the DFT in Mexico City (Fig. 2a). As recently highlighted by Beall et al. (2020), the temperature and the length of the storage can impact the ice nucleation abilities of MA samples. Although, this was not evaluated in the present study, future studies will evaluate how the storage procedure impacts results.*"

**5** line 1616: Again: Umo et al. (2015) looked at only ash particles, so any agreement has to be coincidental, as the n_s for Umo et al. (2015) relates to the surface area of ash particles, only, while your data relate to total atmospheric aerosol with many other particles and with some ash particles, and also with some soot particles in them! This means in detail: When (at a fixed temperature) Umo et al. (2015) report a value of 1000 cm^-2, then this means that on the surface of the ash particles, there are 1000 INP per cm^2 that are active at that temperature. When your BB samples show the same n_s, it means that the BB aerosol has 1000 INP per cm^2 of the OVERALL aerosol. That is clearly something completely different. Check and correct all of your respective comparisons respectively.

A/ We agree with the reviewer that ambient aerosol particles are complex and that their physiochemical properties may differ from those found in pure substance. Following the Editor's advice and to acknowledge this, the following text was added to the revised manuscript. Lines 613-615 "*Note that some of the $n_s$ data in the comparison refer to pure components or aerosol types, while our analysis includes the entire atmospheric aerosol variability, i.e., also the surface of particles not acting as ice nucleating particles.*"

**6** Concerning the derivation of n_s, still more details are needed. If I did not overlook anything, it is nowhere to be found if you used one average surface area from the average size distribution for each of the three phases (MA, BB and AD), and if yes, from which days this average size distribution was constructed (all days for which INP measurements were included?). Or, if the measured size distribution for each separate day was used. At least this information needs to be clearly added. Related is the sentence in line 1498, where you added "(reported by the LasAir)". First, this inserted new information would be better given in the first sentence of the paragraph. (The sentence you inserted it to is more difficult to understand now). And second, the information you give in the review on how you derived the values shown in Fig. 6 should also be given in the manuscript, so that it becomes clear which value you show in that figure.

A/ This section was modified following the reviewer suggestion, as shown above in the answers to the Editor (point #6).

**Minor comments:**

Concerning your method, you answered that direct method comparisons have already been made, citing two papers. Knowing this helps the reader to judge the credibility of your work, and I strongly recommend that you add a short paragraph in which you cite these comparison studies and shortly summarized the respective results for your method.

A/ The following text was added to the revised manuscript. Lines 257-261: "*The results delivered by the Mason et al. (2015) MOUDI-DFT were compared against those reported by the Colorado State University-Continuous Flow Diffusion Chamber (CSU-CFDC). As shown in DeMott et al., (2017), the median INP concentrations from both devices were good agreement.*"

In Lines 421-429 we also briefly described the PICNIC intercomparison campaign where the UNAM-MOUDI-DFT was included. We are not sure if the reviewer is asking us to expand this paragraph further. Given that the detailed information about the PICNIC results will go in a separate manuscript, we do not want to be too detailed here.

**line 1249**: You do not want to describe your method in more details, so as suggested above cite these comparison studies, at least. But still, as readers might want to copy what you do, add at least information on how much does adding this extra plate change in distance in the impactor and as such influence the selected diameters? That may be small but should at least be mentioned, so that others who want to do similar things are aware of the fact that they should use slides as thin as possible.

A/ The following information was added to the revised manuscript. Lines 237-238 "*hydrophobic glass coverslips of 22 mm x 22 mm*"

The MOUDI was traditionally used to collect aerosol particles on different filters to characterize their chemical composition. Each filter (e.g., Teflon, quartz, aluminum, among others) was fixed on each MOUDI stage using a metallic substrate holder provided by the manufacturer. Therefore, we did not add an additional plate to each MOUDI stage. We simply replaced the original substrate holders with new ones that could hold the square glass coverslips.

Finally, as mentioned in our previous answer, additional details about the inter-comparison results were added to the revised manuscript.

**line 1487 ff:** Can it be excluded that these elements come from Saharan dust (during times when this source is weaker or more particles were lost on the way across the Atlantic)? Only then the karstic soil of the region should be taken into consideration. Particularly as you measured close to the ocean, a strong local influence may not be expected.

A/ As explained above, the new Figure 5 shows the enrichment factor of each element. Therefore, we can differentiate between local and external aerosol source.

**line 1555 ff:** You do not want to change these comparisons, which I criticized as being a bit off, and so be it. But at least also mention that the temperatures in air and water are different for the studies done further to the North than for yours, as this may be an important parameter.

A/ Following the reviewer's suggestion, the following text was added to the revised manuscript. Lines 557-559: "*Additionally, given that air temperatures, RH, and sea surface temperatures are considerably different between the Tropics and higher latitudes, these*

*differences could be partially responsible for the latitudinal differences observed in INP concentrations.*"

**line 1577 ff**: As you said above, particles can be lost, and this will likely be the main cause for lower values. This should be repeated here, too, as aging may also contribute, but would show up in a different n_s, rather than in lower concentrations. Concentrations simply HAVE TO BE lower further away from the source.
A/ The following text was added to the revised manuscript. Lines 623-629: "*Given that AD particles travelled >8000 km before reaching the Yucatan Peninsula, the low $n_s$ values calculated for the AD particles are not unexpected. It is likely that the most efficient INPs could have been washed out while travelling over the Atlantic. Also, it is well known that aerosol aging can strongly affect the ice nucleating abilities of mineral dust particles (Kanji et al., 2017). Therefore, the composition of the AD particles that arrived at the Yucatan Peninsula may significantly differ from the AD particles found closer to their source, with different ice nucleation efficiencies.*"

**In your answer (line 810) you say:** "We are not comparing the ice nucleating abilities of the three air masses based on the onset freezing values, we did it based on the n_s values." But actually, the amount of comparison you do for n_s values in Section 3.5 is very limited. There are conclusions given on that in the conclusions section which are not given in Section 3.5, which might merit to be included and discussed there.
A/ The following text was added to the revised manuscript. Lines 631-637: "*Large $n_s$ values can be likely associated with the presence of biological particles as was the case in Si et al. (2018), who linked the high $n_s$ values with the presence of terrestrial biological particles. However, McCluskey et al. (2018) and DeMott et al. (2016) showed that these kind of particles can also be of marine origin as there is large biological activity in marine environments. Rodriguez-Gomez et al. (2020) showed different terrestrial and marine microorganisms that were identified in the Sisal, those of which can be linked with the high ns values found for the MA samples.*"

**Technical issues:**
**line 1345**: Delte the "s" in "INP_s(T)" in equation 3.
A/ Corrected.

**line 1458**: Add ""in the BB season" after "background conditions", as this is not the case during MA.
A/ The text was modified following new Figure 5.

**line 1461**: Use GoM (instead of Gulf of Mexico), as you've done throughout the text.
A/ Corrected.

**line 1470ff:** "… corroborating that the sampled air masses during this season contained a comparatively high mass fraction of particles emitted from BB." At the end of this sentence, it should be added "when PM2.5 was high", as you show yourself that potassium was not always high during this phase (Fig. 5 and Fig. S2a).
A/ The suggested text was added: Line 478.

---

## Author Response (AR3)

We would like to thank the Editor for considering our request. Specific answers and manuscript modifications related to the Editor comments are given below in red text.

**EDITOR**

I am looking forward to the submission of a version of your manuscript in which you either provide a DOI link to a reliable public data repository where your data have been uploaded or, alternatively, attach the data as supplementary information.

A/. The data shown in the figures of our manuscript was added to the Supplementary Material (Tables S3 to S14).

The following text was added to the revised manuscript. Lines 684-685: "***Data availability.*** The Data from the present work can be found in the Supplementary Material (Tables S3-S14). Also, it is available upon request to the corresponding author."